# The tectonic complex regulates membrane protein composition in the photoreceptor cilium

Hanh M. Truong [1], Kevin O. Cruz-Colón[2], Jorge Y. Martínez-Márquez[3], Jason R. Willer [3], Amanda M. Travis[3], Sondip K. Biswas[4], Woo-Kuen Lo [4], Hanno J. Bolz[5,6] & Jillian N. Pearring [3,7] ✉

The primary cilium is a signaling organelle with a unique membrane composition maintained by a diffusional barrier residing at the transition zone. Many transition zone proteins, such as the tectonic complex, are linked to preserving ciliary composition but the mechanism remains unknown. To understand tectonic's role, we generate a photoreceptor-specific *Tctn1* knockout mouse. Loss of Tctn1 results in the absence of the entire tectonic complex and associated MKS proteins yet has minimal effects on the transition zone structure of rod photoreceptors. We find that the protein composition of the photoreceptor cilium is disrupted as non-resident membrane proteins accumulate in the cilium over time, ultimately resulting in photoreceptor degeneration. We further show that fluorescent rhodopsin moves faster through the transition zone in photoreceptors lacking tectonic, which suggests that the tectonic complex acts as a physical barrier to slow down membrane protein diffusion in the photoreceptor transition zone to ensure proper removal of non-resident membrane proteins.

Primary cilia are signaling organelles that relay extracellular cues to the cell through receptors enriched in the ciliary membrane. The ciliary membrane is continuous with the plasma membrane, yet maintains a distinct protein composition due to the presence of a membrane diffusional "gate" that restricts lateral transport between ciliary and non-ciliary membranes[1–6]. Protein enrichment to the ciliary membrane relies on membrane trafficking pathways and ciliary transport carriers that deliver and sort ciliary and non-ciliary components. While previous work has largely focused on how membrane proteins utilize ciliary transport carriers to enter and exit the cilium, the components directly impeding diffusion of membrane proteins remain ambiguous.

The transition zone at the base of the cilium is thought to be where the diffusion barrier resides[1,6]. It is characterized by protein-rich connections between the axoneme and ciliary membrane, called the Y-links, and extracellular membrane densities, called the ciliary necklace[7]. How the transition zone forms a diffusional barrier remains unclear, although it has been postulated that membrane fluidity is reduced due to the abundance of transmembrane proteins and membrane-associated anchors residing within this region.

One membrane-bound complex at the transition zone is the tectonic complex composed of Tctn1, Tctn2, and Tctn3. The tectonics interact with members of the MKS module: Mks1, Tmem216, Tmem67, B9d1, Cc2d2A, and with Cep290, a member of the NPHP module[3,8]. In mice, constitutive knockout of Tctn1, Tctn2, or Tctn3 results in embryonic lethality due to defects in hedgehog signaling[3,8–10]. In cell culture, depletion of tectonics prevents enrichment of ciliary membrane proteins such as polycystin-2, adenylyl cyclase III, Arl13B, and smoothened at the cilium[3,10,11]. In *C. elegans*, loss of Tctn1 does not

[1]Cellular and Molecular Biology Program, University of Michigan, Ann Arbor, MI, USA. [2]Neuroscience Graduate Program, University of Michigan, Ann Arbor, MI, USA. [3]Department of Ophthalmology and Visual Science, University of Michigan, Ann Arbor, MI, USA. [4]Department of Neurobiology, Morehouse School of Medicine, Atlanta, GA, USA. [5]Senckenberg Centre for Human Genetics, Frankfurt am Main, Germany. [6]Institute of Human Genetics, University Hospital of Cologne, Cologne, Germany. [7]Department of Cell and Developmental Biology, University of Michigan, Ann Arbor, MI, USA. ✉e-mail: pearring@umich.edu

affect ciliary GPCR enrichment; however, a non-resident membrane protein, Tram1, was found to be mislocalized into the cilium[11]. While these data highlight that the tectonic complex regulates ciliary membrane composition, whether it plays a direct role in forming the membrane diffusional barrier or an indirect role in regulating active transport of membrane proteins through ciliary transport carriers remains unknown.

Retinal photoreceptors contain a modified primary cilium, called the outer segment, whose ciliary membrane has been elaborated to hold hundreds of disc shaped membranes, called "discs", needed to detect light. The membrane composition of the outer segment is well studied and maintenance of proper outer segment composition is critical for photoreceptor function and health[12–19]. To investigate how tectonic proteins regulate ciliary membrane protein composition, we generated a photoreceptor-specific *Tctn1* knockout mouse. Ciliary outer segment formation and transition zone structure appear normal, however, the entire tectonic complex and several MKS proteins are lost from the transition zone. Over time, non-resident membrane proteins accumulate in the outer segment and rod cells degenerate. The mechanism does not appear to involve alterations in ciliary transport carriers but rather an increase in the rate of diffusion of fluorescent rhodopsin within the transition zone. These results suggest that the tectonic complex may act as a physical barrier within the transition zone that slows diffusion and allows membrane proteins to be properly sorted.

## Results

### Tctn1 is localized to the ciliary transition zone of rod photoreceptor outer segments

Currently, there is no antibody available that detects endogenous mouse Tctn1 despite our and other efforts to generate one. To determine the localization of Tctn1 in photoreceptors, we used in vivo electroporation to express a MYC-tagged full-length Tctn1 (Tctn1-MYC) construct in wild type mouse rod photoreceptors[20]. We co-expressed a rhodopsin-mCherry construct to label the outer segment compartment of transfected rods and stained with either an anti-Cetn1 or anti-Cep290 antibody to label the transition zone in all photoreceptors. Figure 1a shows Tctn1-MYC staining colocalizing with either Cetn1 or Cep290 at the base of rhodopsin-mCherry labeled outer segments. Tctn1-MYC staining was also present within the inner

segment suggesting that overexpression results in some retention within biosynthetic membranes, a common issue when driving low-expressing membrane proteins with the robust rhodopsin promoter[21].

### A rod-specific Tctn1 knockout mouse results in progressive retinal degeneration

To study the functional role of Tctn1 in photoreceptors, we used a CRISPR approach to generate a floxed *Tctn1* allele by inserting LoxP sites flanking exon 2 (Fig. 1b). The *Tctn1^{flox}* allele was verified through Sanger Sequencing of the genomic DNA and genotyped to detect heterozygous or homozygous alleles (Fig. 1c). Mice homozygous for the *Tctn1^{flox}* allele were viable, fertile, and showed no overt phenotypic abnormalities. Cre recombinase expression will cause excision of exon 2 resulting in a frameshift and early termination of *Tctn1*. A rod-specific *Tctn1* knockout mouse (*iCre/Tctn1^{flox}*) was generated by crossing the *Tctn1^{flox}* mice with the iCre75 mouse line in which Cre recombinase expression is driven by the rhodopsin promoter[22]. The rhodopsin promoter begins to express at postnatal day 8 (P8) after initial outer segment ciliogenesis[22,23], so we expect that loss of Tctn1 will occur in mature outer segments.

To assess loss of Tctn1 in our knockout mouse, we first measured *Tctn1* mRNA levels from 1 month retinas and found a significant reduction in the *iCre/Tctn1^{flox}* compared to *iCre/Tctn1^{het}* or *Tctn1^{flox}* controls (Fig. 1d). We also confirmed the excision of exon 2 by Sanger Sequencing the cDNA from *iCre/Tctn1^{flox}* retina (Supplementary Fig. 1). Due to the absence of a Tctn1 antibody, we validated loss of Tctn1 protein by performing tandem mass-tag mass spectrometry (TMT-MS) on isolated outer segments from 3 month *iCre/Tctn1^{flox}, iCre/Tctn1^{het}*, and *C57Bl6/J* wildtype mice. We found that there was no significant difference in Tctn1 peptide abundance between the *iCre/Tctn1^{het}* and *C57Bl6/J* wildtype mice (see Supplementary Data 1) despite a reduction in mRNA levels suggesting that the Tctn1 protein levels are tightly regulated (Fig. 1d). Therefore, we performed most of our analysis between *iCre/Tctn1^{het}* and *iCre/Tctn1^{flox}* littermates. The volcano plot in Fig. 1e shows that Tctn1 peptides along with other members of the tectonic complex including Tctn2, Cep290, B9d1, Cc2d2a, and Mks1 (peptides for Tctn3 were not detected) were significantly reduced, suggesting that the entire tectonic complex is lost from the transition zone in mature *iCre/Tctn1^{flox}* rods.

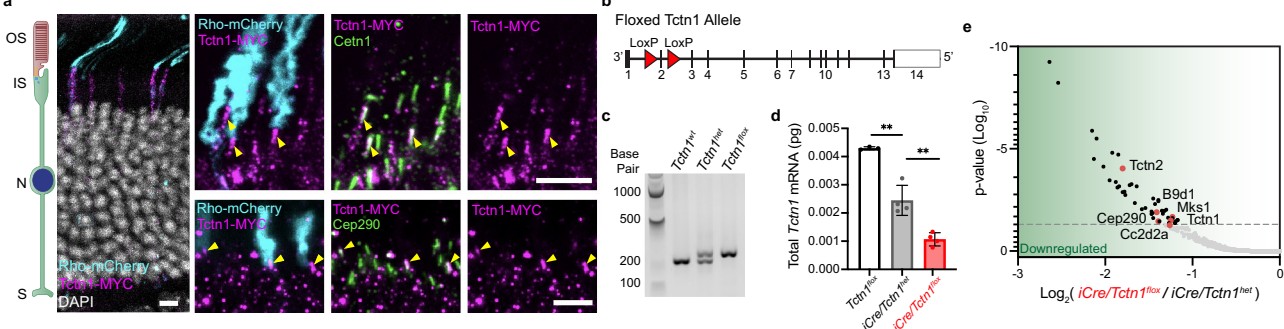

**Fig. 1 | Tctn1 localizes to the transition zone of mouse rod photoreceptors. a** Representative images showing cross-sections of wildtype mouse retinas electroporated with Tctn1-MYC and mCherry-tagged rhodopsin (Rho-mCherry, cyan) stained with anti-MYC (magenta) and anti- Cetn1 (green, top) or anti-Cep290 (green, bottom). Yellow arrowheads indicate expression of MYC-tagged Tctn1 colocalizing with either Cetn1 or Cep290 at the transition zone. Nuclei are counterstained with DAPI (gray). Scale bars, 5 μm. **b** Schematic of the knock-in *Tctn1^{flox}* mouse line that was generated through CRISPR insertion of LoxP sites (red arrowheads) flanking exon 2. **c** Agarose gel showing bands produced by genotyping PCR on isolated genomic DNA from *Tctn1^{wt}, Tctn1^{het}*, and *Tctn1^{flox}* mice. **d** Quantified *Tctn1* mRNA levels obtained through performing qRT-PCR from cDNA isolated from *Tctn1^{flox}, iCre/Tctn1^{het}*, and *iCre/Tctn1^{flox}* 1 month retinas. Each point represents the mean value of a technical triplicate performed on a single biological replicate. Error bars indicate S.D. Statistical significance was determined with an unpaired two-tailed *t* test. ** for *Tctn1^{flox}* and *iCre/Tctn1^{het}*, p = 0.0021; ** for *iCre/Tctn1^{het}* and *iCre/Tctn1^{flox}*, p = 0.0032. **e** Proteins that were significantly downregulated in TMT-MS analysis of purified outer segments from *iCre/Tctn1^{flox}* and *iCre/Tctn1^{het}* 3 month retinas (n = 2 biological replicates). Red dots represent proteins of interest. Volcano plot produced by mapping proteins based on *p* value and log fold change (flox/het) ratio. Proteins that had an adjusted *p* value above 0.05 were considered significant. Source data are provided as a Source Data file. Abbreviations used here and in subsequent figures: outer segment (OS), inner segment (IS), nucleus (N), and synapse (S).

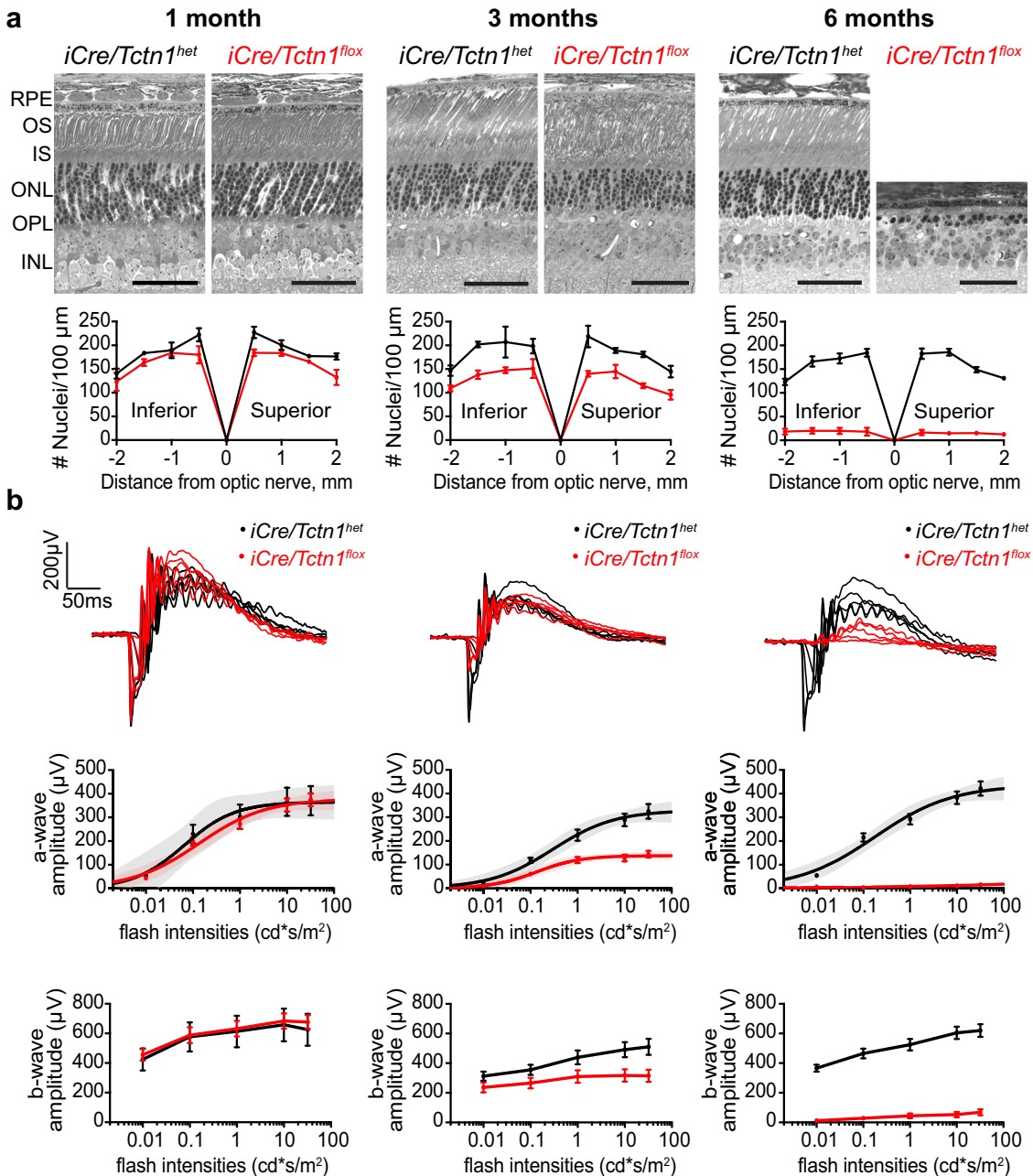

**Fig. 2 | Rod photoreceptors of *iCre/Tctn1^flox^* mice undergo progressive degeneration. a** Semi-thin retinal plastic sections from *iCre/Tctn1^het^* and *iCre/Tctn1^flox^* mice at 1, 3, and 6 month time points show the progressive loss of rods in *iCre/Tctn1^flox^* mice. Scale bars, 50 µm. Abbreviation: retinal pigment epithelium (RPE). Spider diagrams representing the number of nuclei present in the ONL of *iCre/Tctn1^het^* and *iCre/Tctn1^flox^* at 500 µm intervals from inferior to superior. A minimum of *n* = 3 independent animals were analyzed for each genotype at each time point. Error

bars indicate S.D. **b** Representative ERG traces at 1, 3, and 6 months from dark-adapted *iCre/Tctn1^het^* and *iCre/Tctn1^flox^* mice in response to full-field flashes of 0.01, 0.1, 1.0 10, and 32 cd*s/m² (top). The averaged dark-adapted a-wave (middle) and b-wave (bottom) amplitudes for each genotype at each time point are plotted and the a-wave responses are fitted using a single hyperbolic function (middle). *n* = 3 (1 month) or *n* = 4 (3 and 6 month) independent animals were analyzed for each genotype. Error bars indicate S.D. Source data are provided as a Source Data file.

The retinal phenotype of the *iCre/Tctn1^flox^* mouse was initially analyzed in semithin plastic sections and compared to *iCre/Tctn1^het^* controls (Fig. 2a, Supplementary Fig. 2). At 1 month, the *iCre/Tctn1^flox^* mouse had essentially normal retinal sections but displayed progressive photoreceptor degeneration with about one third of cells lost by 3 months. By 6 months, near complete rod degeneration had occurred and only a single layer of cone cells remained. Quantification of rod cell loss is displayed as a spider plot below the histology images. Dark-adapted electroretinography (ERG) analysis corroborated the histological loss of rod photoreceptors, showing a significant reduction in the a- and b-waves starting at 3 months and nearly complete absence by 6 months

in the *iCre/Tctn1^flox^* mice (Fig. 2b). In light-adapted ERGs, cone responses were not significantly different from *iCre/Tctn1^het^* control mice until 6 months when the complete loss of rods likely impacts cone function (Supplementary Fig. 3). Together, these results show that the tectonic complex is needed to maintain photoreceptor cell health.

**Loss of the tectonic complex decreases the abundance of inner scaffold proteins**

We assessed the integrity of the photoreceptor transition zone by immunostaining with Cep290 and Cetn1 antibodies. For this, we isolated fresh retinas from 3 month *iCre/Tctn1^flox^* mice based on Fig. 2. We

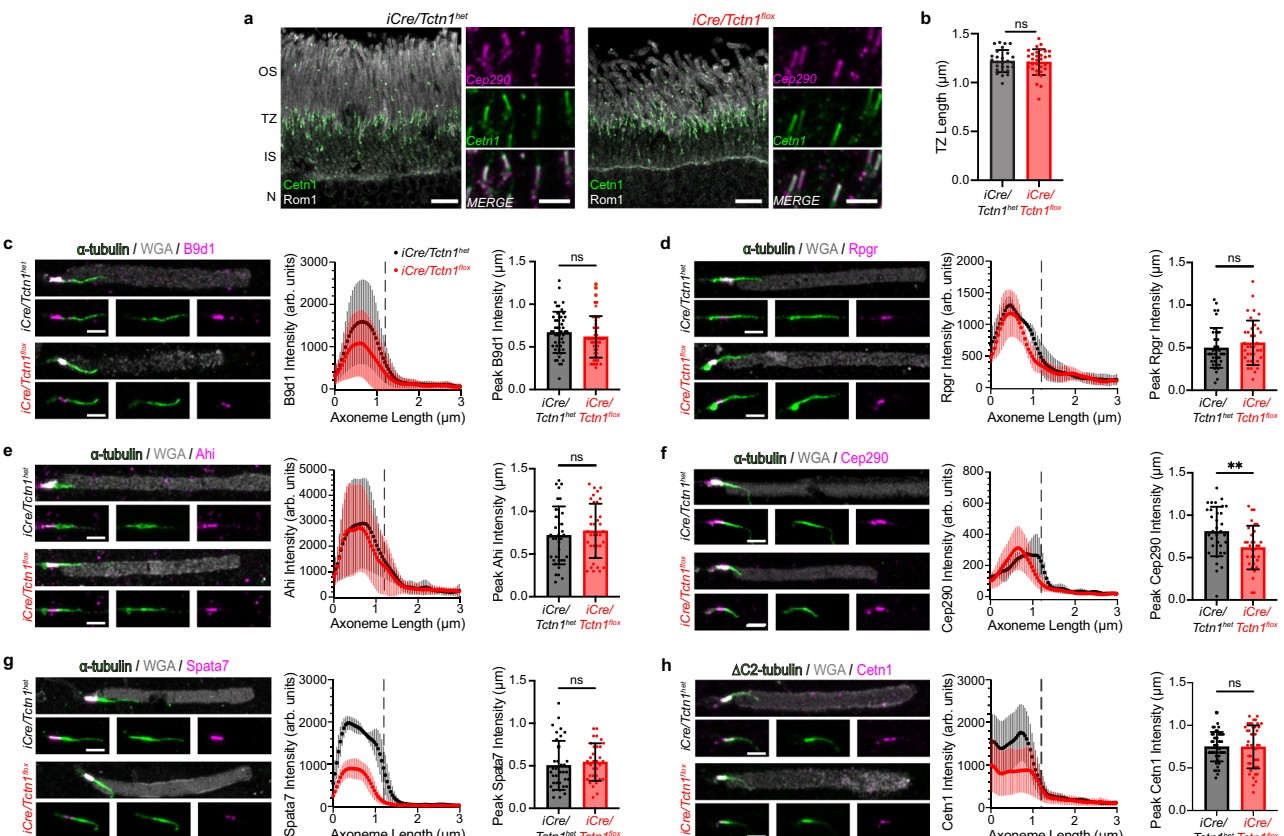

**Fig. 3 | Transition zone staining of *iCre/Tctn1^flox* outer segments.**
**a** Representative images of *iCre/Tctn1^het* and *iCre/Tctn1^flox* 3 month retinas. To the left, confocal images of rod outer segments stained with anti-centrin1 (green), anti-Rom1 (gray). Scale bars, 5 μm. To the right, airyscan images of transition zone stained with anti-centrin1 (green) and anti-Cep290 (magenta). Scale bars, 2 μm.
**b** Bar graph of average transition zone length based on Cep290 staining from *iCre/Tctn1^het* (27 measurements from n = 3 biological replicates) and *iCre/Tctn1^flox* (32 measurements from n = 3 biological replicates). Statistical significance was determined with an unpaired two-tailed t test. ns, p = 0.7448. Error bars indicate S.D. **c–h** Representative airyscan images of isolated outer segments from *iCre/Tctn1^het* and

*iCre/Tctn1^flox* retinas stained with respective transition zone proteins: B9d1, Rpgr, Ahi, Cep290, Spata7, and Cetn1 (magenta), along with α-tubulin or ΔC2-tubulin (green) and WGA (gray). Scale bars, 2 μm. To the right of each set of images, the intensity and localization of each transition zone staining is shown as an averaged intensity plot and bar graph of the position of the fluorescence peak along the axoneme (μm). The intensity of 10–15 transition zones was analyzed for each biological replicate (a minimum of n = 3 for each genotype). Error bars indicate S.D. Statistical significance was determined with an unpaired two-tailed t test. B9d1 ns, p = 0.2892; Rpgr ns, p = 0.2956; Ahi ns, p = 0.5058; Cep290 **, p = 0.0075; Spata7 ns, p = 0.5356; and Cetn1 ns, p = 0.9547. Source data are provided as a Source Data file.

found that both Cep290 and Cetn1 were normally localized at the base of outer segments stained with Rom1, a structural protein at the disc rim (Fig. 3a). Additionally, using the Cep290 staining, we measured the length of the transition zone and did not observe any changes at 3 months even though photoreceptor cell loss is occurring (Fig. 3b).

We then developed a technique to quantify the intensity of transition zone components in *iCre/Tctn1^flox* and *iCre/Tctn1^het* rods. Outer segments were isolated at 3 months, plated onto coverslips, and stained with wheat germ agglutinin (WGA) to label the outer segment membrane, antibodies to α-tubulin or ΔC2-tubulin to label the axoneme, and antibodies to various transition zone proteins. In mouse outer segments, the transition zone region is easily identified by strong enrichment of WGA staining[24]. Using WGA staining, we found that isolated outer segments from *iCre/Tctn1^flox* retinas contained more cellular debris and "broken" outer segments lacking a transition zone compared to *iCre/Tctn1^het* retinas (Supplementary Fig. 4). For analysis of transition zone components, airyscan images were acquired of fully intact outer segments so that line scans could be generated the full length of the axoneme for intensity measurements in each channel. Axonemal intensity measurements from outer segments were aligned to the beginning of the transition zone based on a predetermined threshold of WGA signal marking the outer segment base. Consistent with the TMT-MS analysis (Fig. 1c, see Supplementary Data 1 for full

list), we found that the levels of B9d1, a component of the MKS module and tectonic complex, were reduced in isolated outer segments from *iCre/Tctn1^flox* rods (Fig. 3c) whereas the levels of other transition zone components such as Rpgr and Ahi were not changed in abundance or localization in isolated *iCre/Tctn1^flox* outer segments (Fig. 3d, e). Intensities for Cep290, which had decreased peptide abundance in the TMT-MS data, were unchanged but we did observe a significant shift in the position of the fluorescence peak closer to the basal bodies in *iCre/Tctn1^flox* rods (Fig. 3f). Interestingly, the TMT-MS data also identified a significant reduction in inner scaffold proteins that line the axoneme within the photoreceptor transition zone: Cetn3, Cetn2, Poc5, Spata7, RP1, Rpgrip1L, and Fam161a[25]. We confirmed a reduction of Spata7 and Cetn1 in isolated *iCre/Tctn1^flox* outer segments compared to *iCre/Tctn1^het* controls (Fig. 3g, h). Considering that absence of the tectonic complex reduces these structural components, we next aimed to explore the ultrastructural integrity of *iCre/Tctn1^flox* outer segments.

## Transition zone structure and axoneme remain intact in *iCre/Tctn1^flox* outer segments

We assessed ultrastructural changes of rod outer segments from 3 month *iCre/Tctn1^het* and *iCre/Tctn1^flox* retinal sections using transmission electron microscopy (TEM). We were surprised to find that, despite the loss of tectonic and inner scaffold proteins, the transition

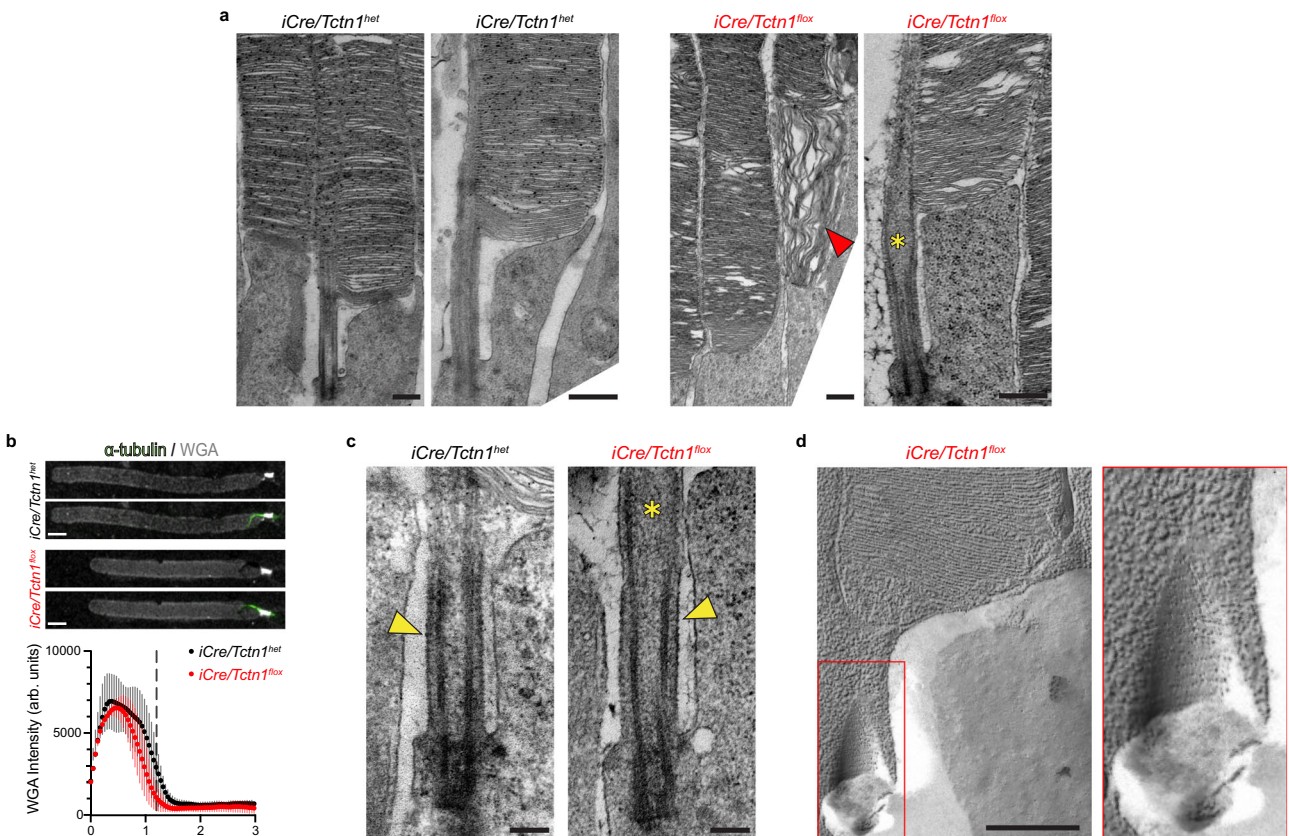

**Fig. 4 | Transition zone and axoneme structure remain intact in *iCre/Tctn1^flox*
outer segments. a** Representative TEM images from cross-sections of 3 month
*iCre/Tctn1^het* and *iCre/Tctn1^flox* retinas. Low magnification to the left; high magnifi-
cation to the right. Red arrowhead marks disorganized discs. Yellow asterisks
indicate bulge in axoneme at distal end of transition zone. Scale bars, 500 nm.
**b** Representative Airyscan images of isolated outer segments from *iCre/Tctn1^het* and
*iCre/Tctn1^flox* retinas stained with α-tubulin (green) and WGA (gray). Scale bar, 2 μm.
Averaged intensity plot is shown below. WGA intensity of 10–15 transition zones
was analyzed for *n* = 3 biological replicates. Error bars indicate S.D. **c** Representative
TEM images of the transition zone from *iCre/Tctn1^het* and *iCre/Tctn1^flox* retinal cross-
sections. Yellow arrowheads indicate the presence of the ciliary necklace present at
the membrane of *iCre/Tctn1^het* and *iCre/Tctn1^flox* outer segments. Scale bar, 200 nm.
**d** Representative freeze fracture TEM images of the transition zone of *iCre/Tctn1^flox*
outer segments. Red box outlines magnification of the transition zone region of
*iCre/Tctn1^flox* outer segments to the right. Scale bars, 500 nm. Source data are
provided as a Source Data file.

zone region appeared normal in the *iCre/Tctn1^flox* retinal sections. One
subtlety we observed was that the microtubule axoneme bulged at the
distal end of the transition zone (Fig. 4a, c). This is reminiscent of the
axoneme splaying observed upon loss of inner scaffold proteins in
photoreceptors[25–28]. The phenotype observed in our 3 month *iCre/
Tctn1^flox* rods is less severe, perhaps due to a slow reduction of the inner
scaffold complex after Cre-mediates loss of Tctn1 at P8. We also found
that the rod outer segments were largely normal in *iCre/Tctn1^flox* sec-
tions, with nicely aligned disc stacks and the presence of a few nascent
discs at the base, however, misaligned and disorganized discs were
observed in ~20% of rod cells (Fig. 4a).

Tctn1 is an extracellular, glycosylated protein and *tctn1* knockout
*Chlamydomonas reinhardtii* were reported to have a disrupted glyco-
calyx on the surface of the flagella membrane[9,29]. To assess the overall
level of glycoproteins at the transition zone in *iCre/Tctn1^flox* rods, we
analyzed the intensity of WGA, a lectin that binds to N-linked carbo-
hydrates on glycoproteins that are highly enriched within the photo-
receptor transition zone. We saw no difference in WGA intensity
present at the transition zone of *iCre/Tctn1^flox* outer segments com-
pared to *iCre/Tctn^het* controls (Fig. 4b). This suggests that the level of
glycoproteins at the transition zone is not altered by the loss of Tctn1.

The membrane-bound nature of the tectonic complex suggests
that it could be a component of the ciliary necklace. However,
electron-dense particles along the transition zone membrane were
observed in *iCre/Tctn1^flox* TEM images (Fig. 4c). To better visualize the

ciliary necklace pattern on the extracellular surface of the membrane,
we performed freeze-fracture electron microscopy and observed the
presence of a "bead-like" necklace around the transition zone mem-
brane at the base of *iCre/Tctn1^flox* rod outer segments (Fig. 4d), sug-
gesting that the tectonic complex does not form the ciliary necklace.
We also did not see any abnormalities in disc morphology or spacing in
*iCre/Tctn1^flox* freeze-fracture images apart from the occasional dying
photoreceptor (Supplementary Fig. 5). Together, these results indicate
that the structural elements of the transition zone remain largely intact
when the tectonic complex is depleted.

**Non-resident membrane proteins accumulate in the ciliary
outer segment when the tectonic complex is absent from rods**
The ciliary transition zone is considered to be a diffusional gate
responsible for maintaining the ciliary membrane protein composi-
tion. The TMT-MS data from isolated outer segments found that many
proteins that are normally excluded from the cilium were significantly
enriched in *iCre/Tctn1^flox* rods (Fig. 5a, Supplementary Data 1). To
investigate this further, we conducted immunofluorescence staining
of the inner segment markers syntaxin3, Snap25, and Na/K ATPase in
*iCre/Tctn1^flox* retinal cross-sections. At 1 month, we found these inner
segment proteins were mislocalized to the outer segment in ~25–40%
of the rods (Supplementary Fig. 6a). By 3 months, we found mis-
localization of inner segment proteins in ~100% of *iCre/Tctn1^flox* rods
(Fig. 5b). The enrichment of syntaxin3 within rod outer segments was

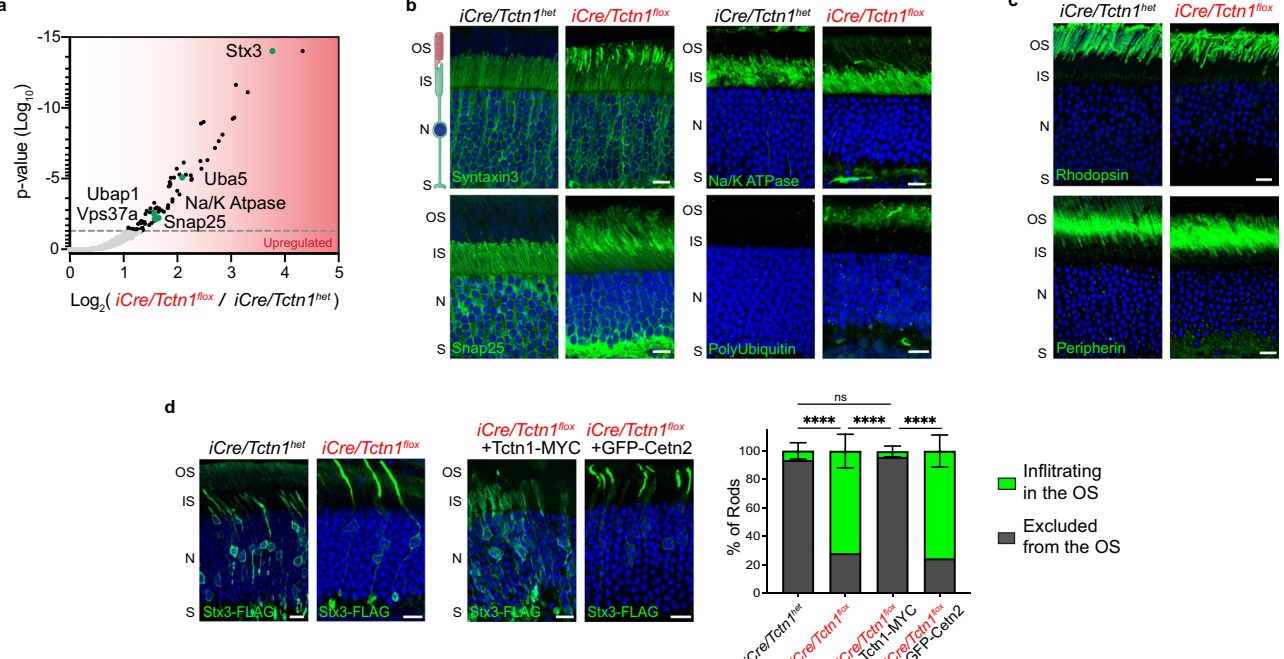

**Fig. 5 | Infiltration of non-resident inner segment proteins into *iCre/Tctn1flox* rod outer segments. a** Proteins that were significantly upregulated in TMT-MS analysis of purified outer segments from *iCre/Tctn1flox* and *iCre/Tctn1het* 3 month retinas (*n* = 2 biological replicates). Green dots represent proteins of interest. Volcano plot produced by mapping proteins based on *p* value and log fold change (flox/het) ratio. Proteins that had an adjusted *p* value above 0.05 were considered significant. **b**, **c** Representative immunofluorescence images of *iCre/Tctn1het* and *iCre/Tctn1flox* retinal cross sections. **b** Inner segment proteins: syntaxin3, snap25, Na/K ATPase, and poly-ubiquitin (green). **c** Outer segment proteins: rhodopsin and peripherin (green). Nuclei are counterstained with DAPI (blue). Scale bars, 10 μm. **d** Images

from 1 month *iCre/Tctn1het* (*n* = 5 biological replicates) and *iCre/Tctn1flox* retinas (*n* = 4 biological replicates) electroporated with Stx3-FLAG or *iCre/Tctn1flox* retinas co-electroporated with Stx3-FLAG and Tctn1-MYC (*n* = 5 biological replicates) or GFP-Cetn2 (*n* = 3 biological replicates). Stx3-FLAG stained with anti-FLAG antibodies (green). Nuclei are counterstained with DAPI (blue). Scale bars, 10 μm. Bar graph on the left shows quantification of the percentage of rods with Stx3-FLAG infiltrating into the outer segment (green) vs excluded from the outer segment (gray). Error bars indicate S.D. Statistical significance was determined with one-way ANOVA. ****, *p* < 0.0001. Source data are provided as a Source Data file.

also shown by immunoprecipitation of syntaxin3 from isolated *iCre/Tctn1flox* rod outer segments (Supplementary Fig. 6b). In contrast to the mislocalization of inner segment proteins, we found that the outer segment proteins rhodopsin, peripherin, Cngβ1, Abca4, and R9AP were normally localized in 3 month *iCre/Tctn1flox* retinas (Fig. 5c, Supplementary Fig. 6c). We did detect a few rods with some rhodopsin staining in the inner segment and the cell body, but this likely represents ongoing photoreceptor degeneration at this age. We also did not see any significant changes in the actin organizing center involved in ongoing disc renewal at the base of the outer segment, further suggesting that disc formation is unaffected (Supplementary Fig. 6c).

A mislocalization of non-ciliary proteins to the ciliary compartment has been previously described for mouse models with defects in the BBSome and in these models, the mislocalization of inner segment proteins was also correlated with an accumulation of poly-ubiquitination within the outer segment[17]. We found that poly-ubiquitination in the outer segment was also elevated in the *iCre/Tctn1flox* retinas (Fig. 5b), suggesting that non-resident membrane proteins are being ubiquitinated. Consistent with elevated ubiquitin, we also observed numerous proteins that have ubiquitin binding domains such as Ubap1, Vps37a, and Uba5 were enriched in the TMT-MS data (Fig. 5a). It is not surprising that the ubiquitin levels were elevated since outer segments do not contain proteosomes, so it cannot clear unwanted membrane proteins from discs. We then analyzed total protein levels for syntaxin3, Snap25, and Na/K ATPase in *iCre/Tctn1het* and *iCre/Tctn1flox* eyecups relative to phosducin, to normalize for photoreceptor cell loss in the degenerating *iCre/Tctn1flox* retina. We found that both syntaxin3 and Snap25 levels were increased in the *iCre/Tctn1flox* eyecups suggesting that the cell is

attempting to compensate for the mislocalization of these proteins (Supplementary Fig. 7).

To test whether the mislocalization of non-resident proteins in *iCre/Tctn1flox* rods could be rescued by re-expressing Tctn1, we performed in vivo electroporations to express a FLAG-tagged syntaxin3 construct (Flag-Stx3) in rod photoreceptors. In *iCre/Tctn1flox* rods, FLAG-Stx3 was mislocalized into the outer segment, however, co-expression with Tctn1-Myc resulted in a strong rescue of FLAG-Stx3 localization back to the inner segment (Fig. 5d). To ensure that this rescue was due to re-inserting the tectonics back into the membrane, we also tested whether co-expressing GFP-Cetn2, a component of the inner scaffold complex, could rescue the FLAG-Stx3 infiltration. Even though Cetn2 was significantly downregulated in our TMT-MS data, we found that its overexpression in *iCre/Tctn1het* rods was unable to restore the barrier as FLAG-Stx3 continued to be predominantly localized to the outer segment (Fig. 5d).

Overall, we find a drastic phenotype in *iCre/Tctn1flox* rods in which the membrane protein composition of the rod outer segment is altered by the accumulation of non-resident membrane proteins. Infiltration of these proteins also corresponds with increased ubiquitin levels and can be rescued by re-expressing Tctn1 in rod cells.

## Loss of the tectonic complex results in increased diffusion of fluorescent rhodopsin through the transition zone

To gain insight into the mechanism by which tectonics regulate outer segment membrane composition, we first tested whether the accumulation of inner segment proteins within *iCre/Tctn1flox* rod outer segments was due to defects in localization or abundance of any of the ciliary transport carriers. Our TMT-MS data showed peptide

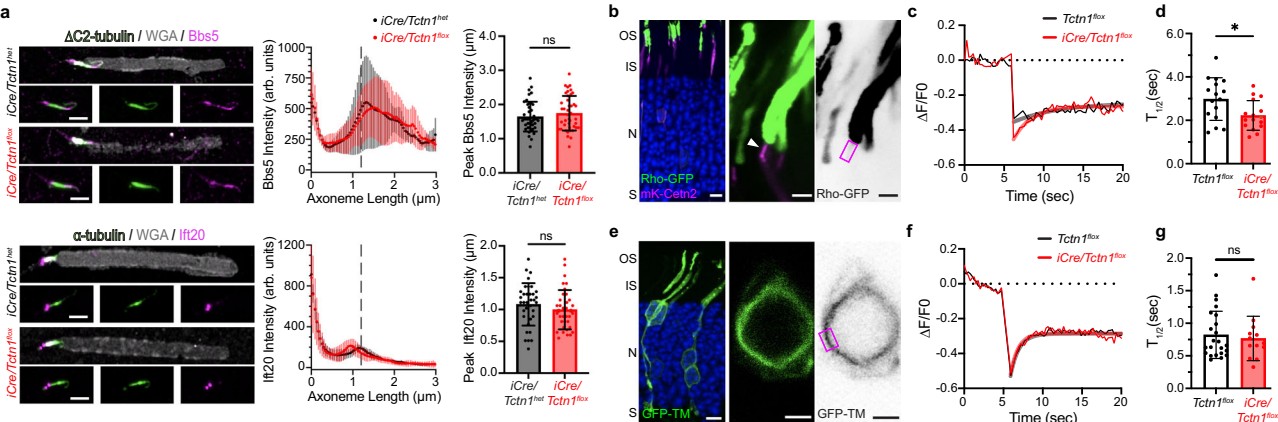

**Fig. 6 | Depletion of Tctn1 causes increased diffusion rates at the transition zone. a** Airyscan images from isolated *iCre/Tctn1^het^* and *iCre/Tctn1^flox^* outer segments stained with anti-Bbs5 or anti-IFT20 antibodies (magenta) along with α-tubulin (green) and WGA (gray). Averaged intensity plots are shown to the right. The intensity of either Bbs5 or IFT20 (10–15 transition zones analyzed per biological replicate, *n* = 3). Error Bars indicate S.D. Statistical significance was determined with an unpaired two-tailed *t* test. BBS5 ns, *p* = 0.3428 and IFT20 ns, *p* = 0.2749.
**b** Representative images from wild type mouse retinas co-electroporated with Rho-GFP (green) and mKate-Cetn2 (magenta). Low magnification to the left (Scale bar, 5 μm); high magnification and contrast images to the right (Scale bars, 2 μm). Magenta box show representative ROI used for photobleaching the transition zone of transfected rods. Nuclei are counterstained with DAPI (blue). **c** Representative FRAP traces from *Tctn1^flox^* and *iCre/Tctn1^flox^* rods plotted for the change in Rho-GFP fluorescence intensity normalized to a pre-bleached average intensity ($\Delta F/F0$) over time in seconds (s). FRAP traces are fitted using a phase I non-linear regression

function. **d** Bar graph plots the average half-time of recovery ($T_{1/2}$, s) calculated from FRAP traces from *Tctn1^flox^* (16 cells, *n* = 5 biological replicates) and *iCre/Tctn1^flox^* (16 cells, *n* = 7 biological replicates) at 1 month. Error bars indicate S.D. Statistical significance was determined with Welch's two-tailed *t* test. *, *p* < 0.018.
**e** Representative image from wild type mouse retinas electroporated with GFP-TM (green). Low magnification to the left (Scale bar, 5 μm); high magnification and contrast images to the right (Scale bar, 2 μm). Magenta box show representative ROI used for photobleaching the plasma membrane of transfected rods.
**f** Representative GFP-TM FRAP traces from *Tctn1^flox^* and *iCre/Tctn1^flox^* rods. **g** Bar graph plots the average $T_{1/2}$ (s) calculated from FRAP traces from *Tctn1^flox^* (23 cells, *n* = 4 biological replicates) and *iCre/Tctn1^flox^* (13 cells, n = 4 biological replicates) at 1 month. Error bars indicate S.D. Statistical significance was determined with Welch's two-tailed t test. ns, *p* = 0.6547. Source data are provided as a Source Data file.

abundances of Tulp1, Tubby, or many BBSome or IFT proteins were not significantly changed in the *iCre/Tctn1^flox^* rods (Supplementary Data 1). We confirmed these data by staining isolated outer segments for either BBS5 or IFT20 and saw no change in their overall localization or fluorescence intensity (Fig. 6a). These data suggest that the presence of ciliary transport carriers is unchanged by the loss of the tectonic complex.

We then tested whether the loss of Tctn1 affects the passive diffusion of transmembrane proteins through the transition zone. A previous study used a rhodopsin-eGFP transgenic mouse to perform fluorescence recovery after photobleaching (FRAP) experiments at the photoreceptor transition zone[30]. To study transition zone diffusion rates, we co-electroporated eGFP-tagged bovine rhodopsin (Rho-GFP) and mKate-tagged centrin2 (mKate-Cetn2) constructs into *iCre/Tctn1^flox^* and *Tctn1^flox^* retinas (Fig. 6b). A 1 μm by 2 μm region of interest (ROI) corresponding to the transition zone was selected using the mKate-Cetn2 signal and the Rho-GFP fluorescence in this ROI was measured before and after photobleaching. Only a modest recovery of Rho-GFP fluorescence was observed (Fig. 6c), likely due to inadvertently photobleaching nascent discs membranes that are the final destination for Rho-GFP. Representative images of GFP fluorescence at baseline, photobleach, and recovery can be found in Supplementary Fig. 8. *Tctn1^flox^* and *iCre/Tctn1^flox^* FRAP traces were modeled to a one-phase non-linear regression function and half-time of recovery ($T_{1/2}$) was calculated. We found that the $T_{1/2}$ was decreased in *iCre/Tctn1^flox^* rods compared to *Tctn1^flox^* controls suggesting an increased rate of diffusion through the transition zone (Fig. 6d).

We then performed FRAP experiments at the plasma membrane to confirm that this change in membrane diffusion was specific to the loss of Tctn1 in the transition zone. We electroporated eGFP-fused to the first transmembrane domain of mGluR1 (GFP-TM), previously shown to have no specific targeting information[31,32]. When expressed into *iCre/Tctn1^flox^* and *Tctn1^flox^* rods the GFP-TM construct localizes throughout all cellular regions (Fig. 6e). For FRAP, an ROI was selected

at the plasma membrane surrounding the nucleus, and the GFP-TM fluorescence was measured before and after photobleaching (Fig. 6e). Recovery of GFP-TM fluorescence did not reach baseline levels, likely due to photobleaching extending beyond the ROI. But by modeling a one-phase non-linear regression function to the recovery curve, we found that the $T_{1/2}$ was unchanged between *Tctn1^flox^* and *iCre/Tctn1^flox^* rods and that membrane diffusion at the plasma membrane was ~3-fold faster than at the transition zone (Fig. 6f, g).

Overall, we find that the loss of the tectonic complex does not impact ciliary transport carriers implying that active transport can still remove non-resident membrane proteins from the outer segment. However, we do find that the mobility of fluorescent rhodopsin is increased in the absence of the tectonic complex suggesting that transmembrane proteins are moving through the transition zone at a higher rate. Therefore, we conclude that the tectonic complex may help form the diffusional barrier by impeding the passive movement of membrane proteins so that they can be efficiently removed by the BBSome (model displayed in Fig. 7).

## Discussion

The ciliary gate residing at the transition zone has a special function to allow the privileged passage of ciliary membrane proteins while restricting the entry of non-resident proteins. In recent years, it has been shown that much of this task is accomplished by active transport carriers that directionally sort membrane proteins into their proper destination[11,17,33]. Our data indicate that active transport works in combination with a barrier residing within the transition zone that slows the lateral diffusion of membrane proteins. By utilizing the light-sensitive cilium of photoreceptor cells, we find that the tectonic complex appears to physically impede the diffusion of membrane proteins through the ciliary transition zone. We postulate that slowing the movement of membrane proteins could allow active transport carriers to sort proteins between ciliary and non-ciliary membranes. In primary cilia, loss of the tectonic complex was shown to cause

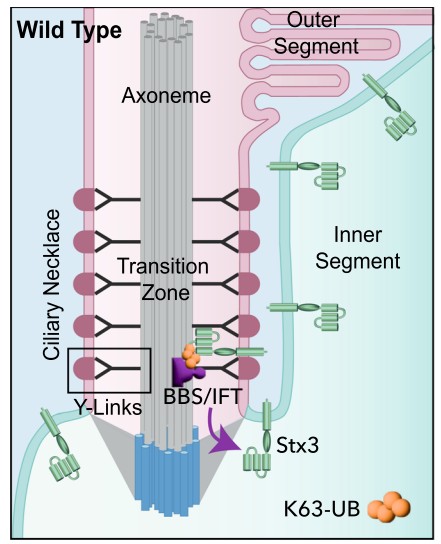
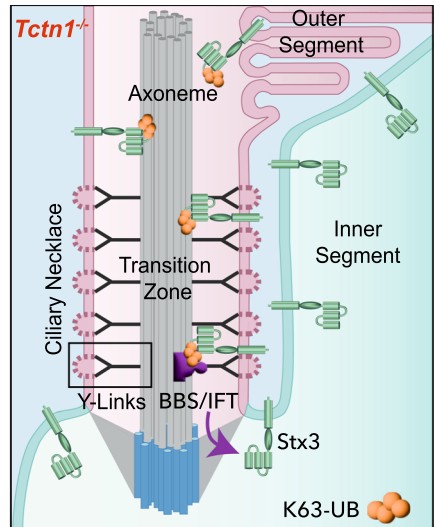
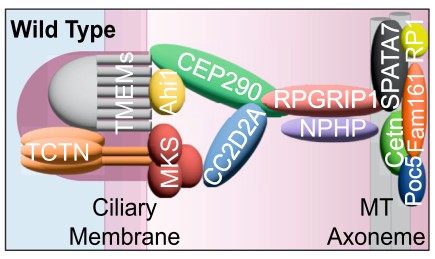
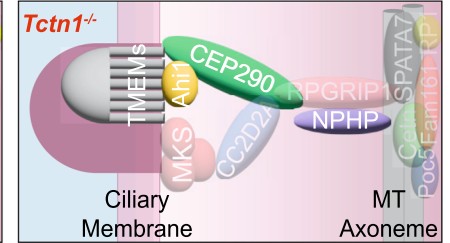

**Fig. 7 | Tectonic resides at the transition zone where it may act as a diffusional gate to ensure proper sorting of membrane proteins.** Model depicting tectonic's proposed role in wildtype photoreceptors to prevent non-ciliary membrane proteins from infiltrating into the ciliary outer segment compartment. In *Tctn1⁻/⁻* photoreceptors, the tectonic complex as well as components of the Mks and inner scaffold complex are downregulated. While this does not lead to gross structural defects in the transition zone or ciliary transport carriers, increased diffusion through the transition zone leads to non-ciliary membrane proteins, such as Stx3, becoming enriched in the enclosed membrane discs of the outer segment over time.

alterations in the membrane protein composition of the cilium[3,8,10]. Changing the signaling profile of the cilium will result in tissue-dependent differences in phenotypes, likely explaining why human mutations in *Tctn* genes are associated with Meckel and Joubert syndromes[8,34–37].

Our results suggest that defects in the diffusional barrier alters outer segment composition; however, our data do not fully rule out that defective active transport may contribute to the phenotype. While localization and expression of ciliary transport carriers remained normal in *iCre/Tctn1ᶠˡᵒˣ* rods, we were unable to assess the activity of these complexes such as loading/unloading of cargos or kinetics of transport. Future studies should be done to determine whether the activity of transport carriers is normal when passive diffusion is altered.

We found that the loss of the tectonics, a heavily glycosylated protein complex, did not disrupt the ciliary necklace or overall glycoprotein staining of the transition zone in rod photoreceptor outer segments. This was surprising as the ciliary necklace is a defining membrane feature of the transition zone previously postulated to function as the barrier. Our TMT-MS data showed that Tmem proteins found at the transition zone were not affected by the loss of tectonics raising the possibility that these proteins could be forming the ciliary necklace structure. The presence of additional membrane complexes within the transition zone is also supported by our FRAP data that showed that rates of diffusion at the plasma membrane were ~3-fold faster than in the transition zone, even in *iCre/Tctn1ᶠˡᵒˣ* outer segments. Moreover, we recognize that while our freeze-fracture TEM provides an overview of the structural organization of the membrane surface, newer techniques such as cryo-EM might provide greater resolution to uncover subtleties within the ciliary necklace of *iCre/Tctn1ᶠˡᵒˣ* outer segments.

Another unexpected finding was that the tectonic complex is helping retain inner scaffold proteins in rod photoreceptors. Inner scaffold proteins, such as Poc5, Fam161, Cetn2, and Cetn3, line the inner wall of the centriole helping to tether the adjacent microtubule triplets together[38]. In photoreceptor outer segments, this inner scaffold complex is also present at the transition zone where it is believed to help tether the adjacent microtubule doublets[25]. Our TMT-MS data revealed a significant decrease in the entire inner scaffold complex in *iCre/Tctn1ᶠˡᵒˣ* outer segments, which we confirmed by immunofluorescence. It is intriguing that the removal of a largely extracellular membrane-bound complex depletes proteins present inside the axoneme and really highlights the interconnected network of transition zone complexes. One potential link between these two seemingly disparate complexes is Rpgrip1, found to be reduced in our MS data set and previously shown to interact with Spata7[39]. Its ortholog found in the primary cilium, Rpgrip1L, interacts with many Mks proteins[40–45] and both Rpgrip1 and Rpgrip1L synergistically regulate transition zone assembly by ensuring enrichment of NPHP and MKS components[46].

Depletion of inner scaffold proteins was previously shown to cause severe microtubule axoneme splaying at the apical side of the photoreceptor transition zone and is associated with retinal degeneration[25–28,47]. TEM images from *iCre/Tctn1ᶠˡᵒˣ* retinal cross-sections did show a subtle bulge of the axoneme at the apical end of the transition zone. While this observation was not present in our wildtype images, a similar bulge can be found in published TEM images of wildtype mice photoreceptors[48]. The bulging axoneme could be due to a reduction in inner scaffold protein, a TEM artifact due to chemical fixation, or a naturally occurring phenomenon due to the presence of actin that is polymerizing to form new discs at this site[49]. Another phenotype reported in the inner scaffold mutant photoreceptors was

the presence of misaligned disc stacks within the outer segments. While we did observe misaligned discs in ~20% of *iCre/Tctn1flox* outer segments, we were unable to determine if this was directly due to the loss of inner scaffold proteins or ongoing rod cell death.

In the vision field, it is often proposed that rhodopsin mislocalization to the rod cell body is a readout for defective ciliary trafficking or gating. Importantly, we found that disrupting the diffusional barrier did not primarily affect the localization of resident outer segment membrane proteins, but instead caused robust mislocalization of inner segment proteins to the outer segment. We believe that this is due to the unique architecture of the photoreceptor outer segment in which membrane proteins that enter become enclosed within the discs making leakage of outer segment targeted proteins back to the inner segment unlikely. Additionally, it was shown that the plasma membrane surrounding the cell body of photoreceptors has the capacity to efficiently clear small amounts of outer segment membrane proteins that leak into the rod cell[50]. The handful of rods that do show rhodopsin staining in the cell body are a better indicator of apoptosis that is occurring during degeneration than a specific trafficking or gating defect, which is clearly observed by syntaxin3 mislocalization in all the rods.

If we consider that rhodopsin molecules are possibly diffusing faster through the transition zone of *iCre/Tctn1flox* rods, it is reasonable to hypothesize that this might alter the rate at which new discs are formed. The rate of disc formation is tightly coupled to the process of outer segment phagocytosis by the RPE to ensure that the length of the outer segment is maintained correctly. For this reason, even if rhodopsin is added more quickly, it is likely that discs are also phagocytosed more quickly. Nevertheless, we believe this is a fascinating point that should be investigated in future studies.

The question then becomes whether the progressive retinal degeneration seen in the *iCre/Tctn1flox* mice is due to either the alteration in the protein composition of the outer segment or the reduction of key housekeeping proteins within the inner segment. Although we observed accumulation of syntaxin3 as early as 1 month of age in the *iCre/Tctn1flox* mice, we did not detect rod cell loss or altered ERG responses at this age. Previous studies have also shown that overexpression of proteins in the outer segment membranes does not result in photoreceptor cell death[51,52], suggesting the outer segment has a high capacity for storing membrane proteins. Together, this suggests that the addition of non-resident membrane proteins within the outer segment does not grossly alter the light-sensing function of rods and is not the primary cause of degeneration. Instead, our TMT-MS data found numerous channels and transporters were enriched in the *iCre/Tctn1flox* outer segments, a place where they cannot function. We found that the photoreceptor has the capacity to accommodate for the misplacement of certain inner segment proteins, such as syntaxin 3 and Snap25, by increasing expression. But, this was not a global response as Na/K ATPase levels were not elevated in the *iCre/Tctn1flox* eyecups; therefore, we believe that the drastic loss of key housekeeping proteins from the inner segment plasma membrane is likely driving the programmed cell death of rods.

## Methods
### Animals
Albino CD-1 wildtype mice used in electroporation experiments were ordered from Charles River Laboratories (Strain # 022; Mattawan, MI). *Tctn1flox* mice were generated by the University of Michigan Transgenic Mouse Core. LoxP sites were inserted upstream and downstream of *Tctn1* exon 2 using Easi-CRISPR targeting strategy in fertilized mouse zygotes via pronuclear microinjections[53]. Manipulated eggs were placed in culture until they developed into blastocysts and then transferred into the oviducts of B6SJLF1/J breeders. Identification of G0 founder chimeras that carried the desired *Tctn1flox* mutation was confirmed by Sanger sequencing. A genotyping assay was designed to

produce a 287 bp product in wild type and a 317 bp product in the *Tctn1flox* allele. A complete list of primers used for sequencing and genotyping can be found in Supplementary Table 1. The G0 founder chimeras were then breed for three generations to *iCre75* mice on a *C57B16/J* background (gift from Dr. Cagri Baseri, University of Michigan) to produce *Tctn1wt/flox* mice for experiments. Rd8 and Rd1 mutations were screened and bred out during these backcrosses.

Mice were handled following protocols approved by the Institutional Animal Care and Use Committees at the University of Michigan (registry number A3114-01). As there are no known gender-specific differences related to vertebrate photoreceptor development and/or function male and female mice were randomly allocated to experimental groups. All mice were housed in a 12/12-h light/dark cycle with free access to food and water.

### qRT-PCR from isolated mouse retinas
A complete list of primers used for qPCR can be found in Supplementary Table 1. Total RNA was extracted from dissected retinas using Trizol (10296010; Invitrogen, Grand Island, NY) using standard manufacturer's recommendations. RNA quality of RIN 7 was confirmed using Agilent High Sensitivity RNA screen Tape. cDNA was synthesized using 500 ng of total RNA per biological sample using iScript Advanced cDNA Synthesis (1725037; Bio-Rad, Hercules, CA). Quantitative PCR was performed on 1 μl of cDNA per biological sample using iTaq Universal SYBR Green Supermix (1725121, Bio-Rad) and a Bio-Rad CFX Real-Time System. Mouse *Tctn1* exon2 was targeted using primers spanning the *Tctn1* exon 2/exon 3 junction. To determine Tctn1 mRNA levels we generated a standard curve of known concentrations of full-length mouse *Tctn1* plasmid (10 ng/μl–0.0001 pg/μl) and approximate *Tctn1* mRNA levels were calculated by plotting the averaged Cq values from each biological samples to the standard curve.

### In vivo electroporation of mouse retinas
DNA constructs were electroporated into the retinas of neonatal mice using the protocol as originally described in ref. 20, but with modifications as detailed in ref. 54. Briefly, P0-P2 mice were anesthetized on ice, had their eyelid and sclera punctured at the periphery of the eye with a 30-gauge needle, and were injected with 0.25–0.5 μL of concentrated plasmid in the subretinal space using a blunt-end 32-gauge Hamilton syringe. Plasmid mixture consists of 2 μg/μL of gene of interest and 1 μg/μL of pRho driving expression of mCherry or RhomCherry as an electroporation marker to identify transfected cells. The positive side of a tweezer-type electrode (BTX, Holliston, MA) was placed over the injected eye and five 100 V pulses of 50 milliseconds were applied using an ECM830 square pulse generator (BTX). Neonates were returned to their mother until collection at P21 or P30.

### DNA constructs
A complete list of primers used for cloning can be found in Supplementary Table 1. For rod-specific expression, cDNA sequences were cloned between 5′ *AgeI* and 3′ *NotI* cloning sites into a vector driven by a 2.2 kb bovine rhodopsin promoter originally cloned from pRho-DsRed, (gift from Dr. Connie Cepko; Addgene plasmid #11156; http://n2t.net/addgene:11156). MYC-tagged Tctn1 was created by cloning the entire mouse Tctn1 sequence from pcDNA-CMV-Tectonic1 (gift from Dr. Elvir Becirovic, University of Munich) followed by a C-terminal MYC sequence. Previously characterized pRho plasmids included: soluble mCherry, mouse rhodopsin C-terminally tagged with mCherry, bovine rhodopsin C-terminally tagged with eGFP, and eGFP-fused to the first transmembrane segment of the metabotropic glutamate receptor 1 (gifts from the Dr. Vadim Arshavsky, Duke University)[31,32,55]. A mKate-Centrin2 was generated by cloning mKate from pCR4-mKate2-K31 (gift from Dr. Vadim Arshavsky, Duke University)[56] in front of the entire mouse *Cetn2* sequence cloned from mouse liver gDNA from a transgenic GFP-Cetn2 mouse (Strain # 027967; Jackson Laboratories, Bar

Harbor, ME)[57]. Full length mouse syntaxin3B was cloned from wildtype mouse retinal cDNA, fused to a C-terminal 3XFLAG sequence, and inserted into pRK plasmid. The pRK plasmid was generated by replacing the bovine rhodopsin promoter from pRho with a human rhodopsin kinase promoter with SV40 SD/SA site cloned from hRK:IZsGreen AAV (a gift from Dr. Tiansen Li)[58].

## Immunofluorescence

**Staining of rod-specific proteins on fixed mouse eyes.** To stain inner segment and outer segment specific proteins, eyes were enucleated at 1 and 3 month time points after sacrificing by $CO_2$ inhalation and drop-fixed in 4% paraformaldehyde in PBS at room temperature for 1–2 h before washing with PBS. Eyecups were dissected and embedded in 4% agarose (BP160-500; Thermo Fisher Scientific, Grand Island, NY) and cut into 100 μm thick sagittal sections with a vibratome (Leica Biosystems, Buffalo Grove, IL). Free-floating sections were blocked in 5% donkey serum (NC0629457, Thermo Fisher Scientific) and 0.5% Triton X-100 in PBS for 1 h at room temperature. Then incubated with primary antibodies overnight at 4 °C, washed with PBS, and then incubated in blocking buffer for 1 h at room temperature with secondary antibodies. Sections were then washed and mounted on slides with Immu-Mount (9990412, Thermo Fisher Scientific) and 1.5 mm coverslips (72204-10; Electron Microscopy Sciences, Hatfield, PA).

**Transition zone staining on mouse retinas.** Immunostaining protocol originally described in ref. 59. Fresh retinas were dissected in Supplemented Mouse Ringers, at pH 7.4 and ~313–320 mOsM [130 mM NaCl, 3.6 mM KCl, 2.4 mM MgCl₂, 1.2 mM CaCl₂, 10 mM HEPES, 0.02 mM EDTA, 20 mM Bicarbonate, 10 mM D-glucose, 1X vitamins, 0.5 mM Na(L)-glutamate, 3 mM Na₂Succinate]. Fresh retinas were blocked in 10% normal donkey serum, 0.3% saponin, 1× cOmplete™ Protease (11836170001; Millipore Sigma; Burlington, MA) diluted in supplemented Mouse Ringers for 2 h at 4 °C. Primary antibodies were diluted in blocking buffer and incubated for 20–22 h at 4 °C and then rinsed and incubated with secondary antibodies for 2 h at 4 °C. Retinas were rinsed and fixed in 4% paraformaldehyde for 30 min at room temperature. Fixed stained retinas were then embedded in 4% agarose (BP160-500, Thermo Fisher Scientific), and cut into 200 μm thick sagittal sections with the vibratome. Sections were mounted with 1.5 mm coverslips (72204-10, EMS) using Prolong Gold mounting media (P36980, Thermo Fisher Scientific).

**Transition zone staining on isolated mouse outer segments.** Immunostaining protocol originally described in ref. 60. To prepare isolated outer segments, retinas were dissected and collected in a microcentrifuge tube containing 225 μL of Mouse Ringers per retina. The retinas were vortexed on HIGH for 1 min and large debris pelleted using a benchtop spinner for 5 s. The supernatant containing isolated outer segments was then plated onto 13 mm poly-L-lysine glass coverslips (354085; Corning, Glendale, AZ) for 5 min before fixation in 4% paraformaldehyde in PBS at room temperature for 5 min. Plated outer segments were then rinsed with PBS and permeabilized for 5 min with an incubation of 0.02% SDS diluted in PBS. Primary antibodies were diluted in 2% donkey serum (NC0629457, Thermo Fisher Scientific) and 0.5% Triton X-100 in PBS and incubated on plated outer segments for either 1 h at room temperature or overnight at 4 °C. Coverslips were then washed and incubated with secondary antibodies for 1–2 h at room temperature before washing and mounting using Prolong Gold mounting media (P36980, Thermo Fisher Scientific).

A complete list of antibodies and recommended dilutions for immunofluorescence can be found in Supplementary Table 2. For all conditions, 10 μg/ml DAPI was added with donkey secondary antibodies conjugated to Alexa Fluor 488, 568, or 647 (Jackson ImmunoResearch; West Grove, PA).

## Fluorescence recovery after photobleaching (FRAP)

Electroporated retinas with GFP expression were carefully dissected in Supplemented Mouse Ringers, at pH 7.4 and ~313–320 mOsM. Fresh retinas were then embedded in 2.5% low-melt agarose (A9045-100G; Sigma-Aldrich, St. Louis, MO) and vibratome sectioned into 200 μm thick sagittal sections. Expressing retinal sections were then transferred to a 35 mm glass bottom petri dish (50-305-807; Ibidi, Fitchburg, WI) and held stationary using a slice anchor with 1.0 mm spacing (64-1417; Warner Instruments, Holliston, MA) fitted to a custom-made plastic holder. The dish containing the expressing retinal sections was then placed in a stage top incubation chamber with temperature, $CO_2$, and humidity control (Carl Zeiss Microscopy, White Plains, NY). 63× objective was used to image Rho-GFP or GFP-TM expressing photoreceptors for FRAP imaging. A 6× zoom was used to focus on a single expressing photoreceptor and either a 2 μm × 1 μm rectangular ROI was generated to target the transition zone region or a 2 μm × 1.5 μm rectangular ROI was generated to target the plasma membrane region for bleaching. The precise time of each cycle varied from ~35–50 ms based on the resolution of the y-axis. All bleaching was performed after 20 cycles using the 405 nm laser at 100% power for 5 cycles. Intensity values were recorded post-bleach for a total of 200 cycles. FRAP measurements were normalized to the average pre-bleach intensity values. Adjusted FRAP measurements were fitted to a log phase-I nonlinear regression curve to determine half recovery times and extents of recovery using Prism 9 (GraphPad Software LLC, San Diego, CA).

## Image analysis

Images were acquired using a Zeiss Observer 7 inverted microscope equipped with a 63× oil-immersion objective (1.40 NA), LSM 800 confocal scanhead outfitted with an Airyscan super-resolution detector controlled by the Ziess Zen v6.0 software (Carl Zeiss Microscopy). Manipulation of images was limited to adjusting the brightness level, image size, rotation, and cropping using FIJI (ImageJ, https://imagej.net/Fiji). All phenotypes were measured using images taken from at least three independent retinas.

**Quantification of outer nuclear layer counts in thin plastic sections.** Thin plastic sections through the optic nerve of *iCre/Tctn1^flox* and *iCre/Tctn1^het* eyecups (n = 4) were collected as described in the Methods Section, Retinal Histology and Transmission Electron Microscopy, and used to quantify the loss of photoreceptor cells. 100 μm² rectangular ROI was generated at 500 μm intervals from the optic nerve. The FIJI cell counter plug-in was used to manually count the number of nuclei in each rectangular ROI. The nuclear counts were graphed on Prism 9 software.

**Quantification of Stx3-FLAG in electroporated mouse retinas.** The FIJI software was used to quantify the subcellular localization of Stx3-FLAG in electroporated retinas of *iCre/Tctn1^flox* and *iCre/Tctn1^het* control mice. Quantification was done by scanning through Z-stack images of the outer nuclear layer and manually counting using the cell counter plugin the presence of 3xFlag-Stx3 exclusively within either the inner segment (counted as "excluded") or the outer segment (counted as "infiltrated") of expressing cells. For each retina, all counted cells from the imaged sections were summed as individual "excluded" and "infiltrated" percentage values. The values from multiple electroporated retina samples were then averaged as the final values for each genotype and treatment.

**Measuring fluorescence intensity of transition zone proteins in isolated outer segments.** For each antibody condition (see Supplementary Table 2 for antibody details), the axoneme was co-stained with either α-tubulin or ΔC2-tubulin antibodies, and outer segment membranes were co-stained with WGA lectin conjugated to Alexa594 (W11262, Thermo Fisher Scientific). Outer segments with intact

transition zones were selected for imaging. For each retina, 10–15 isolated outer segments were imaged using a pre-determined laser power set to the wild type condition for that antibody condition and a 3× axial zoom, to standardize the resolution to 0.04 μm/pixel, with an Airyscan detector on an LSM 800 microscope (Carl Zeiss Microscopy). 0.35 μm Z-stacks were taken to collect the entire thickness of the outer segment, generally 0.9–1.3 μm in depth. Images were stacked, and FIJI software was used to generate a 15-pixel segmented line along the axoneme for fluorescence intensity measurements. Intensity measurements along the line were acquired for every channel so that WGA intensities could be used to define and identify the transition zone within each image. The intensity measurements from each transition zone staining were then aligned to the start of the transition zone, based on predetermined WGA parameters, and plotted using Prism 9 software. From these plots, the area under the curve was calculated to determine the X position along the axoneme (μm) for the fluorescence peak using Prism 9 software. For images with centriole interference, the first 0.4 μm was excluded when calculating the peak.

### Isolation of mouse rod outer segments for proteomic analysis and immunoblotting

Outer segments were isolated from 3 month *iCre/Tctn1^flox^* and *iCre/Tctn1^het^* mice using established techniques with some modifications[60]. Fresh mouse retinas (7 retinas/condition) were dissected and collected in a microcentrifuge tube containing ice-cold 400 μL 16% (w/v) sucrose diluted in Mouse Ringers, at pH 7.4 and ~313–320 mOsM with 1× cOmplete™ Protease (Millipore Sigma; Burlington, MA) and vortexed on HIGH for 1 min. The vortexed solution with retinal debris was then transferred to a 3.5 mL Ultraclear ultracentrifuge tube (Beckman 13 × 51 mm) and was layered on top of ice-cold 900 μL of 27% sucrose and 900 μL 37% sucrose, both diluted in Mouse Ringers with protease inhibitors. The gradient was centrifuged in a swing bucket rotor (SW 50.1) at 50,000 × *g* for 30 min at 4 °C and a purified outer segment fraction could be seen as an opaque band between the 27% and 37% interface. ~600 μL of purified outer segment fraction was carefully collected and transferred to another a 3.5 mL Ultraclear ultracentrifuge tube and was diluted 3× with ice-cold Mouse Ringers with protease inhibitors. The diluted solution was then centrifuged with a fixed-angle rotor (TLA100.3) for 100,000 × *g* for 30 min at 4 °C to pellet the outer segments. The outer segment pellet was gently rinsed with ice-cold Mouse Ringers with protease inhibitors and then resuspended and incubated in 200 μL of hypotonic buffer (10 mM Hepes; pH7.4) for 1 h on ice. The purified outer segment membrane solution was then transferred to a 500 μL Ultraclear ultracentrifuge tube (8 × 34 mm Beckman) and pelleted again by centrifuging in a fixed-angle rotor (TLA120.1) for 100,000 × *g* for 30 min at 4 °C. The supernatant was discarded, and the purified outer segment membrane pellet was resuspended with 30 μL of Buffer A (20 mM Hepes, 100 mM KCl, 2 mM MgCl₂, 0.1 mM EDTA, 0.1% SDS, 0.5% TritonX100; pH 7.4).

### Syntaxin3 immunoprecipitations

Dissected eyecups were sonicated in 250 μL in 1% SDS and 1× cOmplete™ Protease diluted in PBS. The lysates were centrifuged at 16,000 × *g* for 10 min at room temperature and the cleared eyecup protein lysates were used either directly for immunoblotting or for immunoprecipitations. Immunoprecipitations were performed by adding 0.5 μL of Syntaxin3B antibody (15556-1-AP; Proteintech) to either eyecup or outer segment purified protein lysates and incubated at 4 °C rotating overnight. 10 μL Protein A/G magnetic beads (PI88802; Thermo Fisher Scientific) were added to protein lysates and are incubated at 4 °C rotating for 1–2 h. Immunoprecipitants were eluted with 1% SDS in PBS rotating at room temperature for 15 min and processed for immunoblotting.

### Western blotting

Proteins lysates were prepared for immunoblotting. SDS-PAGE using AnykD Criterion TGX Precast Midi Protein Gels (5671124, Bio-Rad) was followed by transfer at 66 mV for 120 min onto Immun-Blot Low Fluorescence PVDF Membrane (1620264, Bio-Rad). Membranes were blocked using Intercept Blocking Buffer (927–70003; LI-COR Biosciences, Omaha, NE). Primary antibodies were diluted in 50%/50% of Intercept/PBST and incubated overnight rotating at 4 °C. The next day, membranes were rinsed three times with PBST before incubating in the corresponding secondary donkey antibodies conjugated with Alexa Fluor 680 or 800 (LiCor Bioscience) in 50%/50%/0.02% of Intercept/PBST/SDS for 2 h at 4 °C. Bands were visualized and quantified using the Odyssey CLx infrared imaging system (LiCor Bioscience). A complete list of antibodies used for Western blot can be found in Supplementary Table 2. Images of the uncropped Western blots can be found in Supplementary Fig. 9.

### Proteomics

**Sample preparation for protein digestion and tandem mass tag (TMT) labeling.** Purified outer segment samples (24.3 μg/condition) were proteolyzed and labeled with TMT 6-plex essentially by following manufacturer's protocol (P190062, Thermo Fisher Scientific). Briefly, upon reduction (5 mM DTT, for 30 min at 45 °C) and alkylation (15 mM 2-chloroacetamide, for 30 min at room temperature) of cysteines, the proteins were precipitated by adding six volumes of ice-cold acetone followed by overnight incubation at −20 °C. The precipitate was spun down, and the pellet was allowed to air dry. The pellet was resuspended in 0.1 M TEAB and overnight (~16 h) digestion with trypsin/Lys-C mix (1:25 protease:protein; Promega) at 37 °C was performed with constant mixing using a thermomixer. The TMT 6-plex reagents were dissolved in 41 μL of anhydrous acetonitrile and labeling was performed by transferring the entire digest to TMT reagent vial and incubating at room temperature for 1 h. The reaction was quenched by adding 8 μL of 5% hydroxyl amine incubating for 15 min. Labeled samples were mixed together, and dried using a vacufuge. An offline fractionation of the combined sample (~200 μg) into eight fractions was performed using high pH reversed-phase peptide fractionation kit according to the manufacturer's protocol (84868, Thermo Fisher Scientific). Fractions were dried and reconstituted in 9 μL of 0.1% formic acid/2% acetonitrile in preparation for LC-MS/MS analysis. Samples with TMT Mass tag were as follows: C57Bl6/J-1 (126), C57Bl6/J-2 (127), i*Cre/Tctn1^het^*–1 (128), i*Cre/Tctn1^het^*–2 (129), i*Cre/Tctn1^flox^*–1 (130), i*Cre/Tctn1^flox^*–2 (131).

**Liquid chromatography-mass spectrometry analysis (LC-multi-notch MS3).** In order to obtain superior quantitation accuracy, we employed multinotch-MS3 which minimizes the precursor ion interference that occurs from fragmentation of co-isolated peptides during MS analysis[62]. Orbitrap Fusion (Thermo Fisher Scientific) and RSLC Ultimate 3000 nano-UPLC (Dionex) was used to acquire the data. 2 μL of the sample was resolved on a PepMap RSLC C18 column (75 μm i.d. × 50 cm; Thermo Fisher Scientific) at the flow-rate of 300 nl/min using 0.1% formic acid/acetonitrile gradient system (2–22% acetonitrile in 150 min; 22–32% acetonitrile in 40 min; 20 min wash at 90% followed by 50 min re-equilibration) and directly spray onto the mass spectrometer using EasySpray source (Thermo Fisher Scientific). Mass spectrometer was set to collect one MS1 scan (Orbitrap; 120 K resolution; AGC target 2 × 10⁵; max IT 100 ms) followed by data-dependent, "Top Speed" (3 s) MS2 scans (collision induced dissociation; ion trap; NCE 35; AGC 5 × 10³; max IT 100 ms). For multinotch-MS3, top 10 precursors from each MS2 were fragmented by HCD followed by Orbitrap analysis (NCE 55; 60 K resolution; AGC 5 × 10⁴; max IT 120 ms, 100–500 m/z scan range). Since the TMT modification is used as a "fixed modification", unlabeled peptides are not identified or used.

**Mass spectrometry data analysis.** Proteome Discoverer (v2.4; Thermo Fisher) was used for data analysis. MS2 spectra were searched against UniProt mouse protein database (Mus muscular ALL_UniProtKB 2021_03) using the following search parameters: MS1 and MS2 tolerance were set to 10 ppm and 0.6 Da, respectively; carbamidomethylation of cysteines (57.02146 Da) and TMT labeling of lysine and N-termini of peptides (229.16293 Da) were considered static modifications; oxidation of methionine (15.9949 Da) and deamidation of asparagine and glutamine (0.98401 Da) were considered variable. Identified proteins and peptides were filtered to retain only those that passed ≤1% FDR threshold. Quantitation was performed using high-quality MS3 spectra (Average signal-to-noise ratio of 6 and <50% isolation interference). A Welch's $t$ test and Benjamin Hochberg correction method was used to generated adjusted $p$ values. A complete list of proteins identified can be found in Supplemental Data 1.

### Electroretinography (ERG)

A Diagnosys Celeris System (Diagnosys, Lowell, MA) was used to perform ERG recordings on the same $n = 4$ $iCre/Tctn1^{het}$ and $n = 4$ $iCre/Tctn1^{flox}$ mice at 1, 3, and 6 months of age. Mice were dark adapted overnight and handled under red light conditions. For each recording, animals received a drop of 0.5% tropicamide to stimulate eye dilation and a drop of 0.5% proparacaine to numb the eyes. The mice were anesthetized using an intraperitoneal injection of ketamine (50 mg/kg bodyweight) and xylazine (5 mg/kg bodyweight). A drop of 0.3% hypromellose (Genteal) is used to cushion the contact with the electrode stimulators. Scotopic responses were recorded for 0.01, 0.1, 1.0 10, and 32 cd s/m² intensity stimuli. Mice were then light adapted for 10 min prior to photopic testing. Photopic responses are recorded sequentially using 10, 32, and 100 cd s/m² stimuli. Photopic flicker responses are then recorded using 20 cd s/m2 cycled at 9.9 Hz. During testing, body temperature is maintained to 37 °C by the built-in heating element.

For ERG analysis, each data point represents the average value and standard error from eight total eyes. A-wave and b-wave values were extracted from the traces using MATLAB 2021b (MathWorks, Natick, MA). Oscillatory potentials were removed by using a 55 Hz low-pass fast Fourier transform filter prior to calculating the a-wave amplitude from the baseline to the trough. The b-wave was calculated from the absolute value of the trough previously calculated for the a-wave to the peak of the response, which was determined after switching to a 10 Hz low-pass filter. The a-wave responses were fit to a single hyperbolic function and plotted using Prism 9 software.

### Retinal histology and transmission electron microscopy

Experiments were performed as originally described in ref. [61] with the following modifications. Animals were deeply anesthetized and perfused for 15 min with 2.5% glutaraldehyde, 2.5% paraformaldehyde in 100 mM cacodylate pH 7.2. The superior eye was marked on the cornea by a low temperature cautery (50-822-500, Fisher Scientific), then enucleated and fixed overnight 4 °C in the same solution. Eyes were washed in 100 mM cacodylate pH7.2 and dissected to eyecups leaving a small notch on the superior side. The eyecups were washed in PBS and post fixed in $OsO_4$ for 1 h. Eyecups washed in $H_2O$ before dehydrated in a series of graded alcohols (50%, 70%, 90%, 95%, 100%) for 10 min each. Eyecups were then incubated in propylene oxide twice for ten minutes each. Eyecups were infiltrated with Epon embedding media by incubating tissue in a solution of 1:3 Epon to propylene oxide, then 1:1, then 3:1 for 1 h each and finally 100% Epon overnight. All infiltration was done while gently rotating tissue in a bench-top rotator. The eyecups were embedded in a beam capsule with 100% Epon and cured in a 60 °C oven overnight. The embedded eyecups were cut through the optic nerve from superior to inferior in 500 nm slices with a glass knife on an Ultracut E ultramicrotome (Leica Biosystems).

Plastic retinal cross-sections were stained with toluidine blue for imaging by light microscopy.

A smaller area of each retinal section that had well-oriented outer segments was then trimmed for further electron microscopy processing. Thin sections (70–90 Å) were collected on the same ultramicrotome with a diATOME diamond knife, placed on cooper grids, and counterstained with 4% uranyl acetate and 3% lead citrate. Electron micrograph images were taken using a JEM 1400 Plus transmission electron microscope (JEOL, Peabody, MA).

### Freeze-fracture electron microscopy

Experiments were performed as originally described in ref. [61] with the following modifications. Eyes were enucleated and drop fixed in 2.5% glutaraldehyde, 2.5% paraformaldehyde in 100 mM cacodylate pH 7.2 at room temperature. Dissected eyecups were washed three times with 0.1 M cacodylate buffer (pH 7.3) and cryoprotected with 25% glycerol in 0.1 M cacodylate buffer. Eyecups were cut into 300-μm slices with a vibratome and sandwiched together on gold-nickel supports and frozen by rapid immersion in Genetron 22 (Honeywell) at near liquid nitrogen temperature. Retinas were fractured at −135 °C in a vacuum of $\sim 2 \times 10^{-7}$ Torr with a Balzers 400 T freeze fracture machine before shadowing at a 45° angle during platinum evaporation to form a platinum replica of about 1.5 nm thickness. The platinum surface was coated with carbon (20 nm) to stabilize it, and the tissue was dissolved with Chlorox. Replicas were washed in distilled water and collected on Girder finder grids, which were imaged at 80 kV on a JEOL 1200EX transmission electron microscope.

### Statistics and reproducibility

For ERG, RNA, western blot, histological, and immunofluorescence analysis $n > 3$ biological replicates were used per condition ($Tctn1^{flox}$, $iCre/Tctn1^{het}$, or $iCre/Tctn1^{flox}$) to ensure reproducibility while reducing animal usage. To control for variability, we designed genetic crosses so that we compared littermates. We observed low variability between biological replicates and phenotypes were apparent and consistent across genotypes. All the results were successfully replicated. All the graphical data are presented as a mean with ±standard deviation. For transition zone staining and FRAP experiments, outliers were determined by applying a robust regression and outlier removal (ROUT, $Q = 10\%$ allowing up to 10% of identified points to be false) before performing a unpaired two-tailed $t$ test on $Tctn1^{flox}$ and $iCre/Tctn1^{flox}$ datasets. For FLAG-Stx3 rescue experiments, a one-way ANOVA was performed. For Western blot analysis, a Welch's two-tailed $t$ test was performed. All statistical analyses were done using Prism 9 software (Graphpad). Statistical test, corresponding $p$ values and the number of biological replicates are listed in the figure legends.

### Reporting summary

Further information on research design is available in the Nature Portfolio Reporting Summary linked to this article.

## Data availability

Source data are provided with this paper. The raw TMT-MS data are provided as an excel sheet (Supplemental Data 1) with this paper and has also been deposited to the ProteomeXchange Consortium via the PRIDE partner repository with the data identifier PXD042270. All other data supporting the findings of this study are available from the corresponding authors upon reasonable request. Source data are provided with this paper.

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

## Acknowledgements

We are grateful to Dr. Kristen Verhey (University of Michigan) and Dr. William Spencer (SUNY Upstate Medical University) for excellent feedback on the manuscript. *Tctn1^{flox}* mice were generated by Dr. Wanda Filipiak and Galina Gavrilina in the University of Michigan Transgenic Animal Model Core part of the Biomedical Research Core Facilities, which is supported by the following NIH P30 grants: CA046592, DK034933, DK081943. The TMT-MS raw data were generated by Dr. Venkatesha Basrur at the Proteomics Resource Facility in the Department of Pathology (University of Michigan). This work used a Leica STELLARIS 8 FALCON Confocal Microscopy System, funded by a National Institutes of Health SIG grant NIH S10OD28612. Eyecup samples for histology and electron microscopy were processed by Brad Nelson, and ERG raw data were collected by Sarah Sheskey all available through the University of Michigan Vision Research Center which is supported by the following NIH P30 grant EY007003. This work was supported by NIH R01 grants EY032491 (J.N.P.) and EY005314 (W.K.L.), NIH T32 grant GM007315 (H.M.T.), Matilda E. Ziegler Research Award (J.N.P.), Career Development Award (J.N.P.) and an Unrestricted Grant to the University of Michigan from Research to Prevent Blindness.

## Author contributions

Conceptualization, H.M.T., H.J.B., and J.N.P.; Methodology, H.M.T. and J.N.P.; Investigation, H.M.T., K.O.CC., J.Y.MM, J.R.W., and A.M.T., and S.K.B; Formal Analysis, H.M.T., J.Y.MM, J.R.W., and A.M.T.; Writing—Original Draft, H.M.T. and J.N.P.; Writing—Review & Editing, H.M.T., K.O.CC., J.Y.M.M, J.R.W., A.M.T., S.K.B, W.K.L., H.J.B., and J.N.P.; Funding acquisition, H.M.T. and J.N.P.; Supervision, J.N.P.

## Competing interests

All authors declare no competing interests.
