## [Peer Review File · Nature Communications]

REVIEWER COMMENTS

Reviewer #1 (Remarks to the Author):

This study assesses the role of the ciliary transition zone (TZ) and MKS module protein, tectonic 1, in photoreceptor outer segment structure, function and molecular composition in mice. Knocking out *Tctn1* specifically in photoreceptors showed that the protein is required for the maintenance, but not the initial establishment, of these cell types. Notably, knockout mice do not show defects in TZ structure, which agrees with previous findings from nematodes (PMID: 26540106; study should be cited). Instead, from immuno-stainings, the mice demonstrate a reduction in the TZ levels of several proteins, including one MKS module protein (B9D1) and two inner scaffold proteins (SPATA7, Cent1), although others (CEP290, AH1, RPGR) appeared unaffected. The authors show some evidence from proteomic analysis of isolated OSs that other TZ and OS proteins are present at reduced levels in the knockout mice (eg. RPGRIP1L, Poc1, Rp1), although these are not confirmed using localisation analyses. The study then shows very nicely that proteins normally not resident in the OS, are ectopically localised to this compartment in the knockout mice. This finding is in agreement with other studies in various systems showing roles for MKS module proteins in excluding non-ciliary proteins from the organelle. Finally, to get at mechanism, the authors show that *Tect1* loss does not affect the OS localisations of two IFT proteins; instead, they find that *Tect1* loss increases the rate of Rhodopsin-GFP trafficking across the TZ of the outer segment. These findings lead to the important conclusion that *Tect1*'s role in ciliary gating does not involve IFT regulation, but rather the regulation of diffusion rates across the TZ membrane.

This study addresses important questions of how the MKS module regulates the ciliary gate. For the most part, the work is well executed, with most observations quantified and appropriately controlled. The findings fit well with other published work linking this module to ciliary gating. Also, the *in vivo* nature of this work is important and must be commended. I think, therefore, that this is a very good piece of work. However, many of the findings, whilst novel in the context of *Tctn1* and the photoreceptor, don't really establish a new understanding of the mechanism by which the MKS module functions. As outlined below, the substantive and most important conclusion from the work about regulation of protein diffusion rates (highlighted in the paper's title, abstract and discussion) is not adequately supported by the presented data. Without more convincing evidence, I feel this work falls short for publication in Nature Communications in its current form.

Major comment:

The conclusion about *Tctn1* regulating protein diffusion across the TZ is based on one set of FRAP experiments, focused on a Rho-GFP reporter (Fig 6B-D). There are several issues in my mind: (1) authors must show a series of images or movies pre and post bleach to show the recovery, (2) I don't see any meaningful Rho-GFP signal in the connecting cilium pre-bleach, which makes me wonder how easy it is to image recovery, especially given issue of photobleaching GFP signals whilst imaging post bleach, (3) how do the authors deal with the very strong signal in the proximal most OS whilst imaging the adjacent TZ signal?, (4) The graph in 6D indicates a T0.5 recovery reduction from 3 to 2 seconds in the mutant, with a p-value of 0.018, which does not instill a lot of confidence. Many of the data points are overlapping, and there is as big outlier in the control sample which would affect the overall mean; what happens to the data if the top and bottom 10% of values are removed from the analysis?, (5) To assess flux of membrane proteins across the TZ, could the authors not bleach entire OS signals and assess OS recovery rates (Chih et al. 2012 used this approach to assess diffusion rates in B9D1 disrupted cells)?, (6) If Rho-GFP is entering the OS at increased rates in the mutant, would we not expect increased levels of Rhodopsin in the OS at steady state? I don't see any such data in the paper, (7) it would be nice to see FRAP data for at least one other OS membrane protein to confirm the Rho-GFP data.

Minor comments:

Line 148: The intensity of CEP290 signal in Fig 3F looks well within the dynamic range for imaging; thus, not sure about the argument provided about how low levels of CEP290 makes it hard to see changes between control and mutant.

Line 166: point out the disorganised disks in the figure.

Line 231/245: The fact that transport carriers localise fine in the mutant does not necessarily mean that cargo loading/unloading is normal. Thus, the authors need to be careful about fully discounting active transport regulation in their model.

Figure 1A: also show just the green channel alone

Reviewer #2 (Remarks to the Author):

Primary cilia are formed by nearly every vertebrate cell and mediate various signalling pathways. Consequently, they ensure the proper development and maintenance of vertebrate organs and structures. In the retina of the eye, a unique primary cilium is present in photoreceptors. Its dysfunction can cause the loss of photoreceptors. Importantly, inherited retinal degeneration due to the loss of photoreceptor cells is a leading cause of human blindness. In this study, Truong et al. analysed the function of Tctn1, a protein localising to the ciliary transition zone. The transition zone functions as the ciliary gatekeeper and Tctn1 is one of the proteins that controls ciliary gating. Truong and colleagues aimed to elucidate the molecular mechanism underlying the regulation of ciliary gating by Tctn1. By investigating mice in which Tctn1 was specifically inactivated in photoreceptors, the authors revealed that Tctn1 deficiency results in the loss of photoreceptors and hence in progressive retinal degeneration. Furthermore, they reported a significant reduction of the tectonic complex components and a decreased abundance of inner ciliary scaffold proteins in isolated Tctn1^{-/-} outer segments. Despite the reduced amount of several ciliary protein in the absence of Tctn1, the structure of the transition zone and the axoneme was not affected. However, a mislocalisation of inner segment proteins to the outer segment in rods lacking Tctn1 was uncovered. By performing electroporation and using FRAP, Truong et al. showed that the half-time of recovery of a rhodopsin-eGFP fusion protein was decreased in the transition zone of Tctn1^{-/-} rods. Based on this finding, the authors suggest that the tectonic complex impedes diffusion of membrane proteins through the transition zone.

The topic analysed in this study is of current importance and of interest to a broad readership. Most of the studies provided in this manuscript were carefully performed and are presented in a comprehensible manner. Nevertheless, there are several points which have to be improved before the manuscript is suitable for publication in Nature Communications:

1) In their work, Truong et al. aimed to "investigate how tectonic proteins regulate ciliary membrane protein composition" (lines 72 and 73). To reach this aim, they "generated a photoreceptor-specific Tctn1 knockout mouse" (lines 73 and 74). Undoubtedly, it is important to uncover the mechanism of how the Tctn proteins control ciliary gating and it is also crucial to investigate the function of proteins in the connecting cilium of photoreceptors. Considering that there is a structural and functional diversity of vertebrate primary cilia and that vertebrates present very specialised cell and tissue-specific types of primary cilia, I feel that the author's statements "We further show that membrane proteins moved faster through the transition zone illustrating that the tectonic complex acts as a physical barrier to slow diffusion of membrane proteins so they can be properly sorted by ciliary transport carriers." (lines 21-24), "Together, these results indicate that the structural elements of the transition zone remain largely intact when the tectonic complex is depleted." (lines 182 and 183), "Therefore, we conclude that the tectonic complex helps to impede passive diffusion of transmembrane proteins, so that they can be efficiently removed by the BBSome." (lines 249 and 250) and "With our findings, we can now appreciate that these ciliary alterations are due to a leaky diffusional barrier." (lines 263 and 264) are overinterpreted since their findings do not allow to draw a general conclusion but only a photoreceptor-specific. As mentioned by Truong and colleagues, "retinal photoreceptors contain a modified primary cilium" (line 69). This implicates that this type of cilium is different from other primary cilia. For this reason, it has to be shown that the findings in photoreceptor cilia are transferable to other types of primary cilia in order to formulate a general statement.

2) In the lines 60 and 61, the authors see Cep290 as a component of the Mks module: "The tectonics interact with six members of the MKS module: Mks1, Tmem216, Tmem67, B9d1, Cc2d2a, and Cep290." However, it was initially found as a member of the Nphp5-6 module (Czarnecki and Shah, 2012, *Trends Cell Biol.*, doi: 10.1016/j.tcb.2012.02.001; Garcia-Gonzalo and Reiter, *J Cell Biol.*, doi: 10.1083/jcb.201111146; Sang et al., 2011, *Cell*, doi: 10.1016/j.cell.2011.04.019) and a functional study confirms the classification in which Cep290 belongs to the Nphp module in vertebrates (Wiegering et al., 2018, *EMBO J.*, doi: 10.15252/embj.201797791).

3) The authors mentioned that they cannot analyse the localisation of endogenous Tctn1 in photoreceptors as "...there is no antibody available that detects endogenous mouse Tctn1..." (line 83). However, previous studies investigated endogenous Tctn1 localisation in mice and in human cells (Garcia-Gonzalo et al., 2011, *Nat Genet.*, doi: 10.1038/ng.891; Vuolo et al., 2018, *Elife*, 2018, doi: 10.7554/eLife.39655; Wiegering et al., 2018, *EMBO J.*, doi: 10.15252/embj.201797791). Due to overexpression effects, the electroporated MYC-tagged full length Tctn1 construct makes it difficult to make a reliable statement about the endogenous localisation of Tctn1 in photoreceptors (Figure 1A). Truong et al. should use the antibody against Tctn1 that has been used in the above mentioned studies. Moreover, the images in Figure 1A are too small. The new images of Figure 1A showing the endogenous Tctn1 photoreceptor localisation should be larger.

4) Lines 107-109 and 113-116: "Due to the absence of a Tctn1 antibody, we validated loss of Tctn1 protein by performing tandem mass-tag mass spectrometry (TMT-MS) on isolated outer segments from 3 month iCre/Tctn1flox, iCre/Tctn1het, and C57Bl6/J wildtype mice. [...] The volcano plot in Figure 1E shows that Tctn1 peptides along with other members of the tectonic complex including Tctn2, Cep290, B9d1, Ccd2da, and Mks1 (peptides for Tctn3 were not detected) were significantly reduced, suggesting that the entire tectonic complex is lost from the transition zone in mature iCre/Tctn1flox rods." If I got it right, isolated outer segments are used for this TMT-MS study. In my opinion, the photoreceptor outer segment contains the transition zone but is not equal to it as it also includes the axoneme. Consequently, this study does not quantify the amount of the mentioned proteins in the transition zone. Potentially, the mentioned proteins might be mislocalised to the axoneme in iCre/Tctn1het rods and the amount of the analysed proteins at the transition zone is unaltered in iCre/Tctn1flox and iCre/Tctn1het rods. To exclude scenarios like this, the authors should perform immunofluorescence-based quantifications at the transition zone to confirm their TMT-MS data. I think the authors tried to do this in Figure 3 but there are two problems. First, how did they define the transition zone as the region of interest in which they quantify protein amounts? By using Cep290 as a reference marker as done in the length measurements of the transition zone? Second, in line with the TMT-MS data, the amount of B9d1 is reduced at the transition zone. However, the transition zone amounts of the other proteins that were shown to be reduced in the TMT-MS study are not shown in Figure 3 (Tctn2, Cc2d2a and Mks1). Are they unchanged?

5) Line 114: Ccd2da should be corrected (Cc2d2a).

6) Lines 191-193: "At 1 month, we found these inner segment proteins were mislocalized to the outer segment in ~25-40% of the rods (Supplemental Figure 5A). By 3 months, we found mislocalization of inner segment proteins in iCre/Tctn1flox rods (Figure 5B)." What is the difference between the situation at 1 month and the situation at 3 months? The percentage?

7) Lines 202-206: "A mislocalization of non-ciliary proteins to the ciliary compartment has been previously described for mouse models with defects in the BBSome and in these models, the mislocalization of inner segment proteins was also correlated with an accumulation of polyubiquitination within the outer segment¹⁷. We found that polyubiquitination in the outer segment was also elevated in the iCre/Tctn1flox retinas (Figure 5B), suggesting that non-resident membrane proteins are being ubiquitinated." The transport of proteins out of the cilium via the BBSome is one process that requires polyubiquitination. Another process that requires polyubiquitination is protein degradation by the ubiquitin-proteasome system. Is proteasomal activity reduced by the loss of Tctn1 in photoreceptors?

8) Lines 279-284: "It is intriguing that removal of a largely extracellular membrane-bound complex depletes proteins present inside the axoneme and really highlights the interconnected network of transition zone complexes. One potential link between these two seemingly disparate complexes is

Rpgrip1, reduced in our data set and previously shown to interact with Spata7, while its ortholog found in the primary cilium, Rpgrip1L, interacts with many Mks proteins³⁷⁻⁴¹." It was shown before that Rpgrip1 is involved in transition zone assembly. In some cell types, Rpgrip1 and Rpgrip1L regulate transition zone assembly synergistically. Both together ensure the proper transition zone amount of e.g. all Tctn proteins in these cell types (Wiegering et al., 2018, EMBO J., doi: 10.15252/embj.201797791). This should be included in the Discussion section.

9) Lines 307-309: "The question then becomes is the progressive retinal degeneration seen in the iCre/Tctn1flox mice due to alteration in the protein composition of the outer segment or reduction of key housekeeping proteins within the inner segment." Is something missing in this sentence?

10) Line 309: "1M" is used for – I guess – 1 month. Sometimes, I found 1 month written out in the text (for example in line 104). The authors should write this consistent throughout the text.

Reviewer #3 (Remarks to the Author):

The transition zone (TZ) is a critical structure of cilia that controls their composition and hence affects their function. The composition and organization of the TZ have been extensively studied, but how the different components of the TZ control cilia entry and affect their composition is not yet fully understood. Here, the authors address this question by focusing on mouse photoreceptors. They developed the first conditional allele of Tctn1 in mouse, a core component of the TZ conserved in many organisms, to inactivate Tctn1 only in photoreceptors. They demonstrate that inactivation of Tctn1, 8 days after birth, leads to progressive retinal degeneration that is almost total at 6 months. Loss of Tctn1 alters TZ composition, as revealed by the loss of the entire tectonic complex and several MKS proteins from the transition zone, but does not noticeably affect TZ ultrastructure as revealed by freeze fracture EM and TEM. Interestingly, loss of Tctn1 affects the recruitment of the inner scaffold complex that could also explain the bulge observed at the distal end of the connecting cilium in Tctn1 deficient photoreceptors. Last the authors show that non-resident proteins are misaddressed in the outer segment of Tctn1 deleted photoreceptors. This defect is not correlated with altered distribution of carriers involved in cilia transport, but based on FRAP experiments of the connecting cilium, the authors propose that passive diffusion is increased inside the cilium in absence of Tctn1. This is a solid and well conducted study with technically challenging experiments. It will be of interest to a wide audience in the cilia field. I only have a small concern regarding the conclusion based on the FRAP experiments as explained below.

Photobleaching experiments (FRAP) of Figure 6B:

In this experiment, what are the arguments to conclude that only passive diffusion is observed during recovery and not both active transport and passive diffusion? The authors' conclusion relies on the quantification of carriers in the outer segment, but it is not clear if this readout is sensitive enough to detect small variations in the rate of transport. Therefore, to show that there are no alterations of active transport inside the connecting cilium, photobleaching experiments should also likely be performed on these carriers, if technically possible.

I am not sure to understand this argument: "Only a modest recovery of Rho-GFP fluorescence was observed (Figure 6C), likely due to inadvertently photobleaching nascent discs that house immobile Rho-GFP".

Minor points:

Figure 3 : quantifications of TZ proteins.

For Cep290 and Ahi1 : even though the curves are indeed reaching the same maximum, they are a bit displaced towards the basal body. In particular for Cep290 which is somehow contradictory with the quantification performed in B? Could this difference be meaningful? Could this explain the difference with TMT-MS data? Please comment.

Figure 4A: iCre/Tctnflox, low magnification panel (left). The connecting cilium is not visible on this panel?

Line 191: "At 1 month, we found these inner segment proteins were mis-localized to the outer segment in ~25-40% of the rods (Supplemental Figure 5A)". Quantifications are not provided on Sup Fig5A, please check correspondence.

Supplemental Figure 5A: please check correspondence between legend and panels (syntaxin 3 and snap25 are apparently not presented?)

Reviewer #4 (Remarks to the Author):

Tctn1 is a component of the tectonic-like complex (TCTN1,2,3) at the ciliary transition zone (TZ). Tctn complex belongs to the TZ's MKS module that also include B9 domain proteins (Mks1, B9d1, B9d2), the coiled-coil proteins Cc2d2a and Cep290, and a few ciliary transmembrane proteins. MKS module has been linked to ciliogenesis and ciliary protein composition modulation. It is widely accepted that MKS module contributes the build of ciliary membrane identity by acting as a diffusion barrier for non-ciliary proteins, but how this is achieved is not unclear. Deletion of Tctn1 in mouse leads to tissue-specific ciliogenesis defect and embryonic lethality due to the disrupted Hedgehog signaling at the developing neural tube. The role of Tctn1 in mammalian photoreceptor cilium has not been explored before.

In this ms, Truong et al investigated the role of Tctn1 in postnatal mouse rods. Using a conditional rod knockout model, they found that deletion of Tctn1 leads to progressive retina degeneration which is obvious by 3 month old. They further identified loss/decrease of other TZ proteins, including MKS module proteins (B9d1, Ahi), axoneme lumen protein Cctn1 and NPHP/ RPGR-interacting protein Spata7. Despite the decrease of TZ proteins, CC ultrastructure (axoneme, Y-link) and ciliary targeting of rod outer segment (OS) proteins (rhodopsin and peripherin2) are unaffected. However, proteome analysis of purified OS found accumulation of a plethora of non-resident proteins at 3 month, including t-SNARE protein syntaxin3 (STX3). This novel finding provided a mechanistic explanation of retina degeneration in Tctn1 KO. Finally, using Rhodopsin-EGFP FRAP, they found that quicker GFP signal recovery in KO than in control. They concluded that deletion of Tctn1 impedes the diffusion barrier at CC, leading to the increased alien membrane protein infiltration into the OS. Overall, this is a well-written paper with nice data illustration and interesting, novel results. However, there are some concerns over the design of several experiments and the data interpretation (see below).

1. The localization of Tctn1: The ms assumed that Tctn1 is an extracellular protein, but previous studies have found that that mouse TCTN1 is not secreted despite it has a signal peptide (Reiter and Skarnes 2006; Garcia-Gonzalo et al 2011). If it's not released, WGA staining to check the glycosylated protein at CC surface (Fig. 4B) has no grounds. The localization of Tctn1 also affects the interpretation of the decrease of other TZ proteins. An intracellular localization is more reasonable here. An immun-TEM analysis of endogenous Tctn1 or transfected Tctn1-Myc can be considered to address this problem. The authors mentioned that an antibody for endogenous mouse Tctn1 is unavailable (line 83), but a recent paper successfully used a commercial antibody for superresolution microscopy analysis of TCTN1 localization at TZ (Conduit, et al 2021).

2. It's unclear why the authors used dissociated rods for staining and quantification of TZ proteins (Fig.3C-H, Fig. 6A). Isn't better to directly use retina sections? The use of vortex method to isolate rod OS (line 470,471) may physically damage the CC membrane, hence causing artifact in staining. Indeed, in Fig.3, section staining of Cctn1 showed no difference between control and KO (Fig.3A), but in isolated rod staining Cctn1 is significantly reduced (Fig. 3H).

3. The authors stated that with deletion of TCTN1, whole tectonic complex is gone (Line 115), but this is only based on the proteomic data. It would be nice to show more evidence, such as TCTN2 immunostaining or TCTN2 electroporation. It would also be nice to discuss how deletion of Tctn1 affects TCTN2/3's ciliary localization.

4. To demonstrate an altered diffusion barrier function of CC membrane, the authors performed FRAP experiment with transfected Rho-EGFP (Fig. 6B-D). There are two concerns about this approach. 1).

The Rho-EGFP expression level may vary between transfected cells. Will the protein expression level affect FRAP kinetics? A Rho-EGFP knockin line (Chan et al 2014) will be a better choice. 2). As it's still controversial if ciliary entry of rhodopin involves IFT, the choice of Rho-EGFP as a marker for passive diffusion is questionable. It would be better to try an unrelated protein, such as GFP fused with STX3 transmembrane domain. According to Baker et al 2008, GFP-STX3TM localizes to the OS in transgenic frog. Without a targeting signal, GFP-STX3TM would be a good marker for passive membrane diffusion.

5. As for the interpretation of accumulation of STX3 and other alien proteins in Tctn1^{-/-} OS, the authors didn't discuss an alternative possibility, i.e. reduced ciliary export of IS proteins by BBSome. It's been reported that non-resident proteins accumulates in OS when BBSome is defective, and this includes STX3 (Datta et al 2015, Hsu et al 2017). The authors showed no change of BBS5 and IFT20 at KO CC (Fig. 6A), but this does not exclude a functional change of BBSome/retrograde IFT. On the other hand, if there is an increased passive diffusion for IS membrane proteins, one would expect that native OS membrane proteins will do so either. This will lead to more OS proteins (rhodopsins, R9AP) moving into the OS, but the author's proteomic data/staining data do not support this notion.

Response to comments raised by the reviewers can be found below in orange font.

Reviewer #1

This study assesses the role of the ciliary transition zone (TZ) and MKS module protein, tectonic 1, in photoreceptor outer segment structure, function and molecular composition in mice. Knocking out *Tctn1* specifically in photoreceptors showed that the protein is required for the maintenance, but not the initial establishment, of these cell types. Notably, knockout mice do not show defects in TZ structure, which agrees with previous findings from nematodes (PMID: 26540106; study should be cited). Instead, from immuno-stainings, the mice demonstrate a reduction in the TZ levels of several proteins, including one MKS module protein (B9D1) and two inner scaffold proteins (SPATA7, Cent1), although others (CEP290, AH1, RPGR) appeared unaffected. The authors show some evidence from proteomic analysis of isolated OSs that other TZ and OS proteins are present at reduced levels in the knockout mice (eg. RPGRIP1L, Poc1, Rp1), although these are not confirmed using localisation analyses. The study then shows very nicely that proteins normally not resident in the OS, are ectopically localised to this compartment in the knockout mice. This finding is in agreement with other studies in various systems showing roles for MKS module proteins in excluding non-ciliary proteins from the organelle. Finally, to get at mechanism, the authors show that *Tect1* loss does not affect the OS localisations of two IFT proteins; instead, they find that *Tect1* loss increases the rate of Rhodopsin-GFP trafficking across the TZ of the outer segment. These findings lead to the important conclusion that *Tect1*'s role in ciliary gating does not involve IFT regulation, but rather the regulation of diffusion rates across the TZ membrane.

This study addresses important questions of how the MKS module regulates the ciliary gate. For the most part, the work is well executed, with most observations quantified and appropriately controlled. The findings fit well with other published work linking this module to ciliary gating. Also, the in vivo nature of this work is important and must be commended. I think, therefore, that this is a very good piece of work. However, many of the findings, whilst novel in the context of *Tctn1* and the photoreceptor, don't really establish a new understanding of the mechanism by which the MKS module functions. As outlined below, the substantive and most important conclusion from the work about regulation of protein diffusion rates (highlighted in the paper's title, abstract and discussion) is not adequately supported by the presented data. Without more convincing evidence, I feel this work falls short for publication in Nature Communications in its current form.

Major comment:

The conclusion about *Tctn1* regulating protein diffusion across the TZ is based on one set of FRAP experiments, focused on a Rho-GFP reporter (Fig 6B-D). There are several issues in my mind:

(1) authors must show a series of images or movies pre and post bleach to show the recovery, **In our resubmission, we have included images showing GFP fluorescence at baseline, bleach, and recovery in Supplemental Figure 8.**

(2) I don't see any meaningful Rho-GFP signal in the connecting cilium pre-bleach, which makes me wonder how easy it is to image recovery, especially given issue of photobleaching GFP signals whilst imaging post bleach,

Yes, the number of Rho-GFP molecules within the TZ compared to those packed into outer segment discs is orders of magnitude different. Performing FRAP within the narrow TZ is difficult but made easier by using the mKate-Cetn2 construct that marks the TZ region. We have included high contrast images in black and white (Figure 6B), to better show the faint GFP signal traversing the TZ.

(3) how do the authors deal with the very strong signal in the proximal most OS whilst imaging the adjacent TZ signal?,

The GFP baseline, bleach, and recovery values are collected from a 1 X 2 μm ROI that we select based on the localization of the mKATE-Cetn2 TZ marker that is co-expressed with Rho-GFP (Figure 6). It is possible that the selected ROIs also contains the very bright Rho-GFP in disc membranes. Rho-GFP entry into disc membranes is a dead end, so after photobleaching there will be no recovery of molecules that have entered disc membranes. This likely contributes to the fluorescence recovery not returning to baseline; however, we are still able to determine diffusion kinetics from the Rho-GFP molecules that do recover as these represent the transient population that is moving through the TZ membrane. In the manuscript, we clarify this point on line 250-252. (Please also refer our response to similar concerns raised by Reviewer 3)

(4) The graph in 6D indicates a T0.5 recovery reduction from 3 to 2 seconds in the mutant, with a p-value of 0.018, which does not instill a lot of confidence. Many of the data points are overlapping, and there is as big outlier in the control sample which would affect the overall mean; what happens to the data if the top and bottom 10% of values are removed from the analysis?,

We applied the ROUT (robust regression and outlier removal) tool on the FRAP data to determine whether any of the T0.5 recovery points could be considered outliers. For the Rho-GFP dataset, we found that even using the most aggressive parameters (Q=10%, which allows for up to 10% of the identified outliers to be false), none of the predicted points were considered outliers suggesting that the data is in fact a Gaussian distribution. Also, when we excluded the top and bottom 10% of the datapoints for Tctn1Flox and Tctn1Flox/iCre, as recommended by the reviewer, we found this enhanced the significance to $**p = 0.0041$ making us even more confident in our conclusion. In our resubmission, we included application of the ROUT tool in the methods, but no other changes were made to this dataset.

(5) To assess flux of membrane proteins across the TZ, could the authors not bleach entire OS signals and assess OS recovery rates (Chih et al. 2012 used this approach to assess diffusion rates in B9D1 disrupted cells)?,

Unlike the primary cilia, the outer segment of rod photoreceptors is composed of tightly stacked enclosed membranes, called “discs”. Photoreceptor discs are enclosed and physically separated from the ciliary membrane, so membrane proteins (e.g. rhodopsin) that are localized in discs are trapped within that membrane domain and unable to be removed from the outer segment until reaching the apical tip where they are phagocytized by the adjacent retinal pigment epithelium (RPE) cells. Outer segment recovery of rhodopsin would take ~10 days and would represent overall renewal of the outer segment compartment, not diffusion through the TZ.

(6) If Rho-GFP is entering the OS at increased rates in the mutant, would we not expect increased levels of Rhodopsin in the OS at steady state? I don't see any such data in the paper,

The outer segment contains hundreds of tightly stacked disc-shaped membranes that are densely packed with rhodopsin molecules. Rhodopsin serves both as a light-sensitive pigment as well as a structural building block needed to properly form these membrane discs. Rhodopsin density with the disc membrane is highly regulated, so rhodopsin expression levels are also tightly regulated. It is not likely that the modestly faster rates of diffusion we see by the loss of tectonic would be enough to elevate rhodopsin levels within a disc. If we consider the TZ is like the road to the outer segment, removal of tectonic would be like taking out some of the speed bumps on that road. While traffic can now flow faster along that road, we don't necessary expect that the number of cars on the road will change. But, if the same number of cars (rhodopsin molecules) are now moving faster, an alternative hypothesis would be that increased rhodopsin diffusion into the outer segment might alter the rate at which discs are formed. The rate of disc formation is tightly coupled to the process of outer segment phagocytosis by the RPE to ensure that the length of the outer segment is maintained correctly. For this

reason, even if rhodopsin is added more quickly, it is likely that discs are also phagocytosed more quickly. The rate of disc formation could be faster in Tctn1 mice, which is a fascinating point (now addressed in the discussion) but requires extensive experimentation and is beyond the scope of this study.

The comment about protein expression levels raised by Reviewer 1 and 3 inspired us to investigate how photoreceptors handle the mislocalization of inner segment proteins. We found that iCre/Tctn1flox retinas had elevated levels of syntaxin3 and Snap25 compared to iCre/Tctn1het when normalized for photoreceptor expression (Supplemental Figure 7). This finding suggests that the photoreceptor accommodates for the functional loss of these proteins (due to sequestration in the outer segment) by increasing expression. This was not a global response, as we saw the Na/K ATPase levels were normal in iCre/Tctn1flox retinas. Together this is further evidence that the loss of inner segment proteins whose expression is not compensated could be driving the degeneration of rods. We have added further discussion regarding this in our revision.

(7) it would be nice to see FRAP data for at least one other OS membrane protein to confirm the Rho-GFP data.

We performed new FRAP experiments using a GFP fused to a single-pass transmembrane domain (GFP-TM) construct expressed in Tctn1flox and iCre/Tctn1flox rods. In mouse photoreceptors, the membrane-rich outer segment serves as a “default” location for any membrane bound protein that does not have specific targeting information (PMID 18981232). The GFP-TM construct was previously expressed in rods and in addition to labeling the plasma membrane surrounding the cell body it was shown to localize to the outer segment (PMID 25009288). We felt that this was a good candidate to use for additional FRAP experiments. As we performed these experiments, we found that the expression of GFP-TM was not nearly as robust as Rho-GFP and the level of fluorescence within the transition zone was so low that we could not reliably photobleach the signal. Despite our best efforts, we were only able to get a single recovery at the TZ from each genotype.

One unexpected outcome from these experiments was that since the GFP-TM construct is expressed in the plasma membrane surrounding the rest of the cell, we were able to perform FRAP at these sites to analyze plasma membrane diffusion in Tctn1flox and iCre/Tctn1flox rods. From these experiments, we found no change in the plasma membrane diffusion rate in the iCre/Tctn1flox rods, supporting a role of Tctn1 specifically in the TZ. Also, we found that membrane proteins moved faster in the plasma membrane compared to the TZ in control Tctn1flox rods, confirming that membrane diffusion is impeded within the TZ. Finally, we found that even when Tctn1 is absent the rate of diffusion through the TZ was not as fast as plasma membrane diffusion, which suggests that additional membrane complexes are present at the Tctn1 deficient TZ to slow membrane diffusion (possibly TMEMs?) and likely forming the ciliary necklace we observed by TEM. While we are disappointed that we could not obtain additional TZ FRAP data during the revision timeframe, we feel that these new FRAP results add important insights to our paper and strengthen our conclusions.

Minor comments:

Line 148: The intensity of CEP290 signal in Fig 3F looks well within the dynamic range for imaging; thus, not sure about the argument provided about how low levels of CEP290 makes it hard to see changes between control and mutant.

We agree with the reviewer. In the revision, we have included additional analysis of protein localization in the TZ by comparing the position of the fluorescence peak along the axoneme (μm) in iCre/Tctn1het vs iCre/Tctn1flox outer segments. This analysis (shown in Figure 3) revealed that nearly all the TZ proteins are normally localized, even if intensity is reduced. One exception was Cep290, which shifted more basally with the loss of Tctn1 even though its intensity levels were unchanged.

Line 166: point out the disorganised disks in the figure.

Done.

Line 231/245: The fact that transport carriers localise fine in the mutant does not necessarily mean that cargo loading/unloading is normal. Thus, the authors need to be careful about fully discounting active transport regulation in their model.

Agreed. We have added an additional section in the discussion to comment on this point.

Figure 1A: also show just the green channel alone

In Figure 1A, we include an individual magenta channel showing Tctn1-myc staining in transfected rod photoreceptors. The green channel is endogenous labeling of Cetn1 or CEP290 at the TZ, and we do not feel it is necessary to show it alone. For better visualization, larger images have been included in the revision.

Reviewer #2

Primary cilia are formed by nearly every vertebrate cell and mediate various signalling pathways. Consequently, they ensure the proper development and maintenance of vertebrate organs and structures. In the retina of the eye, a unique primary cilium is present in photoreceptors. Its dysfunction can cause the loss of photoreceptors. Importantly, inherited retinal degeneration due to the loss of photoreceptor cells is a leading cause of human blindness. In this study, Truong et al. analysed the function of Tctn1, a protein localising to the ciliary transition zone. The transition zone functions as the ciliary gatekeeper and Tctn1 is one of the proteins that controls ciliary gating. Truong and colleagues aimed to elucidate the molecular mechanism underlying the regulation of ciliary gating by Tctn1. By investigating mice in which Tctn1 was specifically inactivated in photoreceptors, the authors revealed that Tctn1 deficiency results in the loss of photoreceptors and hence in progressive retinal degeneration. Furthermore, they reported a significant reduction of the tectonic complex components and a decreased abundance of inner ciliary scaffold proteins in isolated Tctn1^{-/-} outer segments. Despite the reduced amount of several ciliary protein in the absence of Tctn1, the structure of the transition zone and the axoneme was not affected. However, a mislocalisation of inner segment proteins to the outer segment in rods lacking Tctn1 was uncovered. By performing electroporation and using FRAP, Truong et al. showed that the half-time of recovery of a rhodopsin-eGFP fusion protein was decreased in the transition zone of Tctn1^{-/-} rods. Based on this finding, the authors suggest that the tectonic complex impedes diffusion of membrane proteins through the transition zone.

The topic analysed in this study is of current importance and of interest to a broad readership. Most of the studies provided in this manuscript were carefully performed and are presented in a comprehensible manner. Nevertheless, there are several points which have to be improved before the manuscript is suitable for publication in Nature Communications:

1) In their work, Truong et al. aimed to “investigate how tectonic proteins regulate ciliary membrane protein composition” (lines 72 and 73). To reach this aim, they “generated a photoreceptor-specific Tctn1 knockout mouse” (lines 73 and 74). Undoubtedly, it is important to uncover the mechanism of how the Tctn proteins control ciliary gating and it is also crucial to investigate the function of proteins in the connecting cilium of photoreceptors. Considering that there is a structural and functional diversity of vertebrate primary cilia and that vertebrates present very specialised cell and tissue-specific types of primary cilia, I feel that the author’s statements:

“We further show that membrane proteins moved faster through the transition zone illustrating that the tectonic complex acts as a physical barrier to slow diffusion of membrane proteins so they can be properly sorted by ciliary transport carriers.” (lines 21-24),

“Together, these results indicate that the structural elements of the transition zone remain largely intact when the tectonic complex is depleted.” (lines 182 and 183),

“Therefore, we conclude that the tectonic complex helps to impede passive diffusion of transmembrane proteins, so that they can be efficiently removed by the BBSome.” (lines 249 and 250) and

“With our findings, we can now appreciate that these ciliary alterations are due to a leaky diffusional barrier.” (lines 263 and 264) are overinterpreted since their findings do not allow to draw a general conclusion but only a photoreceptor-specific. As mentioned by Truong and colleagues, “retinal photoreceptors contain a modified primary cilium” (line 69). This implicates that this type of cilium is different from other primary cilia. For this reason, it has to be shown that the findings in photoreceptor cilia are transferable to other types of primary cilia in order to formulate a general statement.

The role for tectonic and MKS proteins has been well studied in other types of primary cilium (PMIDs: 21763481, 21725307, 26540106, 21422230, 25869670, 22179047, 26392567, 26595381). By building off a rich history of research, we do not feel that we are overstating our findings as these proteins were first theorized to play a role in forming a membrane gate through loss-of-function studies that showed alterations in the ciliary membrane composition. FRAP experiments also found that loss of either B9d1 (a MKS complex protein) or septin2 (a cytoskeleton protein required for B9d1 localization, PMID 20558667) result in increased recovery of membrane proteins in the cilium. However, due to the limited resolution of the TZ in primary cilium these studies did not look directly at passive diffusion in the TZ membrane, so had to infer that recovery at ciliary membrane was faster due to alteration in the TZ. One unique feature of the photoreceptor cilium – extended TZ length – allows us to directly test passive diffusion within this membrane.

As to Reviewer 2’s comment that “their findings do not allow to draw a general conclusion but only a photoreceptor-specific”, we feel that this is an unfair statement to reduce the impact of our work as “specialized” when the cilia field is dominated by model organisms such C elegans, Chlamy, and cultured cells all of which contain cilia with specialized functions. Moreover, human genetics and proteomic data suggest that the composition of the BB, TZ, INV, and axoneme are highly conserved between the primary and photoreceptor cilium.

2) In the lines 60 and 61, the authors see Cep290 as a component of the Mks module: “The tectonics interact with six members of the MKS module: Mks1, Tmem216, Tmem67, B9d1, Cc2d2a, and Cep290.” However, it was initially found as a member of the Nphp5-6 module (Czarnecki and Shah, 2012, Trends Cell Biol., doi: 10.1016/j.tcb.2012.02.001; Garcia-Gonzalo and Reiter, J Cell Biol., doi: 10.1083/jcb.201111146; Sang et al., 2011, Cell, doi: 10.1016/j.cell.2011.04.019) and a functional study confirms the classification in which Cep290 belongs to the Nphp module in vertebrates (Wiegering et al., 2018, EMBO J., doi: 10.15252/embj.201797791).

Thank you for catching this, we have corrected our oversimplified description.

3) The authors mentioned that they cannot analyse the localisation of endogenous Tctn1 in photoreceptors as “...there is no antibody available that detects endogenous mouse Tctn1...” (line 83). However, previous studies investigated endogenous Tctn1 localisation in mice and in human cells (Garcia-Gonzalo et al., 2011, Nat Genet., doi: 10.1038/ng.891; Vuolo et al., 2018, Elife. 2018, doi: 10.7554/eLife.39655; Wiegering et al., 2018, EMBO J., doi: 10.15252/embj.201797791). Due to overexpression effects, the electroporated MYC-tagged full length Tctn1 construct makes it difficult to make a reliable statement about the endogenous localisation of Tctn1 in photoreceptors (Figure 1A). Truong et al. should use the antibody against Tctn1 that has been used in the above mentioned studies. Moreover, the images in Figure 1A are too small. The new images of Figure 1A showing the endogenous Tctn1 photoreceptor localisation should be larger.

The literature referenced all use the same commercial Tctn1 antibody from Proteintech (15004-1-AP) to stain human cell lines (NIH3T3 cells or hTERT-RPE). Only the Wiegeling et al., 2018, EMBO J paper show Tctn1 staining in a mouse cell line (MEFs); however, in this paper staining specificity was NOT confirmed with a knockout control. It should be noted that the Garcia-Gonzalo et al., 2011 paper generated *Tctn1*^{-/-} MEFs, but never showed Tctn1 staining was absent in this cell line and did not use the commercial antibody to stain MEFs, only hTERT-RPEs.

We purchased this antibody and tested its ability to detect endogenous Tctn1 in mouse retinas. First using Western blot, we observed a band at ~50 kDa from retinal lysates, but this band was not reduced in *iCre/Tctn1*^{fllox} retinas, so is likely non-specific (See below Panel A). This ~50 kDa band is also not likely to be Tctn1 as it was not glycosylated (Panel B). Expression of the myc-tagged mouse Tctn1 in HEK293T cells does show that the Tctn1 protein is glycosylated (Panel C). It should be noted that Tctn1 is predicted to be ~65 kDa and the Tctn1-myc band runs as expected at ~70 kDa. We then tested and found that the commercial Tctn1 antibody did not recognize the expressed Tctn1-myc protein in HEK293T by Western blot (Panel D). Finally, we used the Tctn1 Proteintech antibody on retinal sections expressing Tctn1-myc in rod photoreceptors (intracellular “ER” expression is observed in this image) and again found the Proteintech antibody did not recognize mouse Tctn1-myc expressed in tissue (Panel E).

From these results, we concluded that the commercial Tctn1 antibody from Proteintech does not recognize mouse Tctn1. We attempted to generate our own antibody from a mouse Tctn1 peptide but after 2 injections ended up with similar non-specific results.

Proteintech-EndogenousTctn1 antibody does not recognize mouse Tctn1

(A) *Tctn1*^{fllox} and *iCre/Tctn1*^{fllox} retinal lysates immunoblotted with antibodies against either Proteintech-Tctn1 or Phosducin. (B) Deglycosylation assay of wildtype retinal lysates immunoblotted with antibodies against Proteintech-Tctn1 or rhodopsin. (C) Deglycosylation assay of AD293 cell lysates untransfected or expressing myc-tagged Tctn1 immunoblotted with antibodies against Myc or Hsp90. (D) AD293 cell lysates untransfected or expressing myc-tagged Tctn1 immunoblotted with antibodies against Proteintech-Tctn1, Myc, or Hsp90. (E) Representative images showing cross-sections of wildtype mouse retinas electroporated with Tctn1-MYC stained with anti-MYC (green) and anti- Tctn1 (magenta). Nuclei are counterstained with DAPI (blue). Scale bars, 5 μm

4) Lines 107-109 and 113-116: “Due to the absence of a Tctn1 antibody, we validated loss of Tctn1 protein by performing tandem mass-tag mass spectrometry (TMT-MS) on isolated outer segments from 3 month *iCre/Tctn1*^{fllox}, *iCre/Tctn1*^{het}, and C57Bl6/J wildtype mice. [...] The volcano plot in Figure 1E shows that Tctn1 peptides along with other members of the tectonic complex including Tctn2, Cep290,

B9d1, Ccd2da, and Mks1 (peptides for Tctn3 were not detected) were significantly reduced, suggesting that the entire tectonic complex is lost from the transition zone in mature iCre/Tctn1flox rods." If I got it right, isolated outer segments are used for this TMT-MS study.

In my opinion, the photoreceptor outer segment contains the transition zone but is not equal to it as it also includes the axoneme. Consequently, this study does not quantify the amount of the mentioned proteins in the transition zone. Potentially, the mentioned proteins might be mislocalized to the axoneme in iCre/Tctn1het rods and the amount of the analysed proteins at the transition zone is unaltered in iCre/Tctn1flox and iCre/Tctn1het rods. To exclude scenarios like this, the authors should perform immunofluorescence-based quantifications at the transition zone to confirm their TMT-MS data. I think the authors tried to do this in Figure 3 but there are two problems. First, how did they define the transition zone as the region of interest in which they quantify protein amounts? By using Cep290 as a reference marker as done in the length measurements of the transition zone? Second, in line with the TMT-MS data, the amount of B9d1 is reduced at the transition zone. However, the transition zone amounts of the other proteins that were shown to be reduced in the TMT-MS study are not shown in Figure 3 (Tctn2, Cc2d2a and Mks1). Are they unchanged?

The reviewer is correct, our TMT-MS data was performed on isolated outer segments. As the reviewer suggests we confirmed protein reduction and assessed localization of candidates using immunofluorescence-based quantifications from isolated outer segments (Figure 3). In isolated outer segments, the TZ was defined by WGA staining, previously reported to be enriched within the TZ of the outer segment (Besharse et al 1985, PMID: 3156973). Bright WGA staining at the TZ is clearly observed in all our images (Figure 3) as well as shown in Figure 4 where WGA fluorescence intensity is plotted. We have included additional language in the Methods section to be clear about how the analysis was conducted. In Figure 3, we have also included additional analysis to quantitatively assess localization of the signal along the axoneme. This analysis revealed that Cep290 is more basally displaced within the TZ of iCre/Tctn1flox outer segments.

To be fully transparent, we have also now included zoomed out images of isolated outer segments stained with WGA from both iCre/Tctn1het and iCre/Tctn1flox retinas in Supplemental Figure 4. We find that there is a reduced number of intact outer segments in the iCre/Tctn1flox; however, we only perform the intensity analysis on outer segments with intact transition zones. By biasing our data toward the more structurally complete outer segments we may be underestimating the reduction of some of the TZ components. But, whether the fragile TZ found in the iCre/Tctn1flox preparations is directly due to loss of tectonic and Mks proteins or indirectly due to instability of outer segments due to retinal degeneration, remains unknown.

Tctn2 and Mks1 are commercially available antibodies and were tested on isolated outer segment but did not yield reliable and specific staining at the TZ. The Cc2d2a antibody was made in house by the Reiter lab (PMID: 21725307) and was not available for use in our study.

5) Line 114: Ccd2da should be corrected (Cc2d2a).

Done

6) Lines 191-193: "At 1 month, we found these inner segment proteins were mislocalized to the outer segment in ~25-40% of the rods (Supplemental Figure 5A). By 3 months, we found mislocalization of inner segment proteins in iCre/Tctn1flox rods (Figure 5B)." What is the difference between the situation at 1 month and the situation at 3 months? The percentage?

Yes, the number of rod photoreceptors that have the phenotype is different. We have now clarified in the text that by 3 months of age we find all the rod cells contain mislocalized inner segment proteins. So, from ~40% of rods at 1 month compared to ~100% of rods at 3 month.

7) Lines 202-206: “A mislocalization of non-ciliary proteins to the ciliary compartment has been previously described for mouse models with defects in the BBSome and in these models, the mislocalization of inner segment proteins was also correlated with an accumulation of polyubiquitination within the outer segment¹⁷. We found that polyubiquitination in the outer segment was also elevated in the iCre/Tctn1flox retinas (Figure 5B), suggesting that non-resident membrane proteins are being ubiquitinated.” The transport of proteins out of the cilium via the BBSome is one process that requires polyubiquitination. Another process that requires polyubiquitination is protein degradation by the ubiquitin-proteasome system. Is proteasomal activity reduced by the loss of Tctn1 in photoreceptors? **The photoreceptor outer segment is a ciliary compartment that does not contain any proteasomes. Ubiquitinated proteins are generally excluded from the outer segment, as seen in wild type mice (Figure 5). Once a membrane protein enters the outer segment discs it is trapped. The accumulation of ubiquitinated signal within this compartment suggests that these proteins are not properly being removed before they enter the membrane discs (disc formation occurs apical of the TZ).**

8) Lines 279-284: “It is intriguing that removal of a largely extracellular membrane-bound complex depletes proteins present inside the axoneme and really highlights the interconnected network of transition zone complexes. One potential link between these two seemingly disparate complexes is Rpgrip1, reduced in our data set and previously shown to interact with Spata7, while its ortholog found in the primary cilium, Rpgrip1L, interacts with many Mks proteins³⁷⁻⁴¹.” It was shown before that Rpgrip1 is involved in transition zone assembly. In some cell types, Rpgrip1 and Rpgrip1L regulate transition zone assembly synergistically. Both together ensure the proper transition zone amount of e.g. all Tctn proteins in these cell types (Wiegering et al., 2018, EMBO J., doi: 10.15252/embj.201797791). This should be included in the Discussion section.

Thank you for the suggestion. We included more information about Rpgrip1L in the discussion (lines 305-307).

9) Lines 307-309: “The question then becomes is the progressive retinal degeneration seen in the iCre/Tctn1flox mice due to alteration in the protein composition of the outer segment or reduction of key housekeeping proteins within the inner segment.” Is something missing in this sentence?

Thank you, we have edited this sentence for clarity.

10) Line 309: “1M” is used for – I guess – 1 month. Sometimes, I found 1 month written out in the text (for example in line 104). The authors should write this consistent throughout the text.

Thank you for the suggestion. We made all the “month” terminology consistent.

Reviewer #3

The transition zone (TZ) is a critical structure of cilia that controls their composition and hence affects their function. The composition and organization of the TZ have been extensively studied, but how the different components of the TZ control cilia entry and affect their composition is not yet fully understood. Here, the authors address this question by focusing on mouse photoreceptors. They developed the first conditional allele of Tctn1 in mouse, a core component of the TZ conserved in many organisms, to inactivate Tctn1 only in photoreceptors. They demonstrate that inactivation of Tctn1, 8 days after birth, leads to progressive retinal degeneration that is almost total at 6 months. Loss of Tctn1 alters TZ composition, as revealed by the loss of the entire tectonic complex and several MKS proteins from the transition zone, but does not noticeably affect TZ ultrastructure as revealed by freeze fracture EM and TEM. Interestingly, loss of Tctn1 affects the recruitment of the inner scaffold complex that could also explain the bulge observed at the distal end of the connecting cilium in Tctn1 deficient photoreceptors. Last the authors show that non-resident proteins are misaddressed in the outer segment of Tctn1 deleted photoreceptors. This defect is not correlated with altered distribution of carriers involved in cilia transport, but based on FRAP experiments of the connecting cilium, the authors propose that passive

diffusion is increased inside the cilium in absence of Tctn1. This a solid and well conducted study with technically challenging experiments. It will be of interest to a wide audience in the cilia field. I only have a small concern regarding the conclusion based on the FRAP experiments as explained below.

Photobleaching experiments (FRAP) of Figure 6B:

In this experiment, what are the arguments to conclude that only passive diffusion is observed during recovery and not both active transport and passive diffusion? The authors' conclusion relies on the quantification of carriers in the outer segment, but it is not clear if this readout is sensitive enough to detect small variations in the rate of transport. Therefore, to show that there is no alterations of active transport inside the connecting cilium, photobleaching experiments should also likely be performed on these carriers, if technically possible.

Thank you for your comment. We also agree and have added a new section in the discussion commenting on active transport carriers, which while we find to be normally localized and expressed could still be contributing to the phenotype. Ideally, we would be able to image IFT or BBSome trains in the photoreceptors TZ. This is a very challenging experiment to perform and something that has yet to be accomplished in the field.

I am not sure to understand this argument: "Only a modest recovery of Rho-GFP fluorescence was observed (Figure 6C), likely due to inadvertently photobleaching nascent discs that house immobile Rho-GFP".

We now recognize that "immobile Rho-GFP" is probably not well explained especially since even in disc membranes rhodopsin is highly diffusible. The comment refers to the fact that even though we select a specific mKate-Cetn2 marked TZ region, we are likely inadvertently photobleaching Rho-GFP present in disc membranes distal to the TZ (Please see the image in which we have highlighted with the red arrow the presence of brighter Rho-GFP fluorescence in the proximal outer segment). This region represents Rho-GFP that has entered nascent disc membranes that are formed apical to the TZ. The density of rhodopsin within disc membranes is high and is these membranes are believed to act as a binding sink to drive dense packing (PMID: 33614665). With this in mind, we consider that Rho-GFP entry into disc membranes is a

dead end, so we would not see recovery of these molecules after photobleaching. The fraction of Rho-GFP that is in discs likely contributing to the modest fluorescence recovery that we observe for Rho-GFP that does not return to baseline in the TZ FRAP experiments. We thank the reviewer for catching this imprecise statement have altered the text in this section to be clearer.

Minor points:

Figure 3 : quantifications of TZ proteins.

For Cep290 and Ahi1 : even though the curves are indeed reaching the same maximum, they are a bit displaced towards the basal body. In particular for Cep290 which is somehow contradictory with the quantification performed in B? Could this difference be meaningful? Could this explain the difference with TMT-MS data? Please comment.

We appreciate the reviewer's sharp eye. This comment led us to further analyze protein localization in isolated outer segments by determining position of the fluorescence peak along the axoneme (μm) in iCre/Tctn1het vs iCre/Tctn1flox outer segments. This analysis (shown in Figure 3) revealed that even though the intensity levels were unchanged, Cep290 was shifted more basally with the loss of Tctn1. All other TZ proteins maintained their normal localization.

Figure 4A: iCre/Tctnflox, low magnification panel (left). The connecting cilium is not visible on this panel?

Correct, but the connecting cilium is shown in the right panel. The left panels highlight disorganized discs observed next to a normal outer segment stack in the Tctn1 KO mouse. We have now labelled disorganized discs with a red arrowhead.

Line 191: "At 1 month, we found these inner segment proteins were mis-localized to the outer segment in ~25-40% of the rods (Supplemental Figure 5A)". Quantifications are not provided on Sup Fig5A, please check correspondence.

These are approximate values based on qualitative assessments, therefore, are not quantified.

Supplemental Figure 5A: please check correspondence between legend and panels (syntaxin 3 and snap25 are apparently not presented?)

Only syntaxin 3 staining at 1 month was provided for reference.

Reviewer #4:

Tctn1 is a component of the tectonic-like complex (TCTN1,2,3) at the ciliary transition zone (TZ). Tctn complex belongs to the TZ's MKS module that also include B9 domain proteins (Mks1, B9d1, B9d2), the coiled-coil proteins Cc2d2a and Cep290, and a few ciliary transmembrane proteins. MKS module has been linked to ciliogenesis and ciliary protein composition modulation. It is widely accepted that MKS module contributes the build of ciliary membrane identity by acting as a diffusion barrier for non-ciliary proteins, but how this is achieved is not unclear. Deletion of Tctn1 in mouse leads to tissue-specific ciliogenesis defect and embryonic lethality due to the disrupted Hedgehog signaling at the developing neural tube. The role of Tctn1 in mammalian photoreceptor cilium has not been explored before.

In this ms, Truong et al investigated the role of Tctn1 in postnatal mouse rods. Using a conditional rod knockout model, they found that deletion of Tctn1 leads to progressive retina degeneration which is obvious by 3 month old. They further identified loss/decrease of other TZ proteins, including MKS module proteins (B9d1, Ahi), axoneme lumen protein Cctn1 and NPHP/RPGR-interacting protein Spata7. Despite the decrease of TZ proteins, CC ultrastructure (axoneme, Y-link) and ciliary targeting of rod outer segment (OS) proteins (rhodopsin and peripherin2) are unaffected. However, proteome analysis of purified OS found accumulation of a plethora of non-resident proteins at 3 month, including t-SNARE protein syntaxin3 (STX3). This novel finding provided a mechanistic explanation of retina degeneration in Tctn1 KO. Finally, using Rhodopsin-EGFP FRAP, they found that quicker GFP signal recovery in KO than in control. They concluded that deletion of Tctn1 impedes the diffusion barrier at CC, leading to the increased alien membrane protein infiltration into the OS. Overall, this is a well-written paper with nice data illustration and interesting, novel results. However, there are some concerns over the design of several experiments and the data interpretation (see below).

1. The localization of Tctn1: The ms assumed that Tctn1 is an extracellular protein, but previous studies have found that that mouse TCTN1 is not secreted despite it has a signal peptide (Reiter and Skarnes 2006; Garcia-Gonzalo et al 2011). If it's not released, WGA staining to check the glycosylated protein at CC surface (Fig. 4B) has no grounds. The localization of Tctn1 also affects the interpretation of the decrease of other TZ proteins. An intracellular localization is more reasonable here. An immuno-TEM analysis of endogenous Tctn1 or transfected Tctn1-Myc can be considered to address this problem. The authors mentioned that an antibody for endogenous mouse Tctn1 is unavailable (line 83), but a recent paper successfully used a commercial antibody for superresolution microscopy analysis of TCTN1 localization at TZ (Conduit, et al 2021).

Supplemental Figure 2 from Reiter and Skarnes 2006 (PMID: 16357211) showed that Tctn1 has a functional signal peptide that when appended to alkaline phosphatase results in secretion from Cos7 cells. Full length Tctn1 does not appear to be secreted, but this is likely due to its binding to Tctn2 and

Tctn3 (single pass transmembrane proteins). Further confirmation that Tctn1 is a secretory protein can be found in our own data and data from Garcia-Gonzalo et al 2011 (PMID: 21725307). We both find that Tctn1 is N-glycosylated and when overexpressed in cell culture or photoreceptors, Tctn1 is largely “trapped” within the ER with a small fraction reaching the TZ. A dual localization pattern between intracellular membranes and target destination is common when overexpressing secretory proteins. For Tctn1, it is likely that stoichiometric binding to Tctn2 and Tctn3 ensures proper trafficking to the TZ, so some of the overexpressed Tctn1 remains “trapped” in the ER. Importantly, using syntaxin 3 infiltration in the outer segment as a read out for function, we find that the fraction of Tctn1 that does reach the TZ is sufficient to rescue the diffusional gate in iCre/Tctn1flox rods. Importantly, syntaxin 3 localization is not restored when we express Cetn2, an intracellular component that is downregulated in the MS data, further suggesting that tectonic is acting as the gate. Finally, we analyzed WGA staining in the TZ after a recent paper from Chlamy showed that the “glycocalyx” surrounding the TZ membrane was disrupted by the loss of Tctn1 (PMID: 35810181). Immunogold labelling of Tctn1-HA in this paper showed that it resided at the membrane.

Please refer to our response to Reviewer 1 as Conduit et al 2021 also used the same Tctn1 Proteintech antibody on hTERT-RPE1 cells. As shown above, the Proteintech commercial antibody was produced against a human TCTN1 and does not recognize mouse Tctn1.

2. It's unclear why the authors used dissociated rods for staining and quantification of TZ proteins (Fig.3C-H, Fig. 6A). Isn't better to directly use retina sections? The use of vortex method to isolate rod OS (line 470,471) may physically damage the CC membrane, hence causing artifact in staining. Indeed, in Fig.3, section staining of Cetn1 showed no difference between control and KO (Fig.3A), but in isolated rod staining Cetn1 is significantly reduced (Fig. 3H).

The reasons are largely technical in nature. For retinal sections, the only way to reliably stain the transition zone is to use fresh tissue. Fresh retinas are stained with antibodies, fixed, embedded in agarose, and cut on a vibratome into ~200 μm thick sections. We found that antibody penetration across the retina was variable and imaging the TZ region in the thick sections to be inconsistent, even in wild type retinal sections. Therefore, we could not quantify any fluorescent intensities when staining for transition zone proteins on fresh retinal sections. The images were selected to show as representatives of TZ length. We chose to measure TZ length from the fresh retina Cep290 staining to ensure that physically isolating the outer segments did not affect these values. Figure 3A images are not controlled for intensity.

To look at intensity levels, we then turned to staining isolated outer segments, which we found to be highly reproducible for several TZ antibodies. This preparation also allowed us to develop an image analysis pipeline to examine the fluorescent intensity of antibody staining across the axoneme. In the revision, we take this one step further by analyzing the position of the fluorescence peak along the axoneme (μm). This analysis (shown in Figure 3) revealed that Cep290 was shifted more basally with the loss of Tctn1. Overall, we believe that this new method is very powerful and will be used by other vision science labs in the future.

3. The authors stated that with deletion of TCTN1, whole tectonic complex is gone (Line 115), but this is only based on the proteomic data. It would be nice to show more evidence, such as TCTN2 immunostaining or TCTN2 electroporation. It would also be nice to discuss how deletion of Tctn1 affects TCTN2/3's ciliary localization.

We agree and these experiments were attempted. The commercial Tctn2 antibody did not work for immunofluorescence on isolated outer segments, and we were unable to get exogenous Tctn2-HA to localize to the TZ in wildtype rod photoreceptors. All the Tctn2-HA expression was “trapped” in the ER. We were unable to resolve these technical issues before publication but feel that our conclusion that

the entire tectonic complex is disrupted by loss of Tctn1 is also supported by previous interaction and genetic complementation studies (PMID: 28846093, 26540106, 21725307, 28800946).

4. To demonstrate an altered diffusion barrier function of CC membrane, the authors performed FRAP experiment with transfected Rho-EGFP (Fig. 6B-D). There are two concerns about this approach. 1). The Rho-EGFP expression level may vary between transfected cells. Will the protein expression level affect FRAP kinetics? A Rho-EGFP knockin line (Chan et al 2014) will be a better choice. 2). As it's still controversial if ciliary entry of rhodopsin involves IFT, the choice of Rho-EGFP as a marker for passive diffusion is questionable. It would be better to try an unrelated protein, such as GFP fused with STX3 transmembrane domain. According to Baker et al 2008, GFP-STX3TM localizes to the OS in transgenic frog. Without a targeting signal, GFP-STX3TM would be a good marker for passive membrane diffusion.

Yes, expression matters as the number of molecules traversing the TZ at a given time appears to be very few. We found that we were only able to get FRAP recordings from rods with very bright Rho-GFP expression. 1) In electroporation experiments, expression levels are controlled by injecting the same concentration of Rho-GFP in Tctn1^{flox} and iCre/Tctn1^{flox} retinas. The suggested human Rho-EGFP knockout mouse line has rod degeneration, which could complicate experiments when crossed onto the iCre/Tctn1^{flox} mice that also has rod cell loss overtime. Furthermore, we feel that trying to obtain FRAP recordings without the mKate2-Cetn2 TZ marker as a guide would be nearly impossible. 2) During resubmission, we attempted to get more TZ FRAP using a GFP-fused to mGluR1 TM (also shown in Baker et al 2008 to have no targeting signal). However, the expression level of the GFP-TM was much lower than Rho-GFP and so we did not get reliable photobleach within the TZ. The untargeted nature of the GFP-TM construct did allow us to get FRAP recordings from the plasma membrane, now included in Figure 6. (A detailed response also provided for Reviewer 1)

5. As for the interpretation of accumulation of STX3 and other alien proteins in Tctn1^{-/-} OS, the authors didn't discuss an alternative possibility, i.e. reduced ciliary export of IS proteins by BBSome. It's been reported that non-resident proteins accumulates in OS when BBSome is defective, and this includes STX3 (Datta et al 2015, Hsu et al 2017). The authors showed no change of BBS5 and IFT20 at KO CC (Fig. 6A), but this does not exclude a functional change of BBSome/retrograde IFT. On the other hand, if there is an increased passive diffusion for IS membrane proteins, one would expect that native OS membrane proteins will do so either. This will lead to more OS proteins (rhodopsins, R9AP) moving into the OS, but the author's proteomic data/staining data do not support this notion.

This was also brought up by other reviewers. We agree and have mentioned in the discussion that we have not accessed the activity of active transport carriers. Ideally, we would be able to image IFT or BBSome trains in the TZ of photoreceptors, but this is very challenging and something that has yet to be accomplished in the field.

A similar concern regarding rhodopsin protein levels within the outer segment was raised by Reviewer 1. Please refer to our response there. Thank you.

REVIEWERS' COMMENTS

Reviewer #1 (Remarks to the Author):

I appreciate the efforts the authors have made to improve the FRAP data, which is very important for the manuscript's conclusion that the tectonic complex establishes a barrier that impedes diffusion of membrane proteins across the connecting cilium. For example, the authors tried experiments with a different OS membrane protein reporter (eg. GFP-TM); unfortunately, the signal intensity was too low at the photoreceptor CC to generate enough data. The data they did generate with the GFP-TM at the plasma membrane, whilst in line with the notion of the tectonic complex impeding diffusion at the CC, is circumstantial at best. The representative CC images of pre- and post-bleach Rho-GFP in the new Supplemental Figure 8 (note: this figure is incorrectly referred to as Fig S7 in the manuscript) is nice to see; however, it is very hard for me to be convinced that the level of signal recovery is sufficient to make strong conclusions. Also, I wonder why the control trace does not drop to the same extent as the iCre/Tect-flox trace immediately post bleach? One of the reviewers suggested assessing the diffusion barrier model in another system. I don't think this was unreasonable given the marginal nature of the data in Figure 6 (in my opinion), and the weight put on that data to derive a major conclusion. Thus, in the final reckoning, I remain unconvinced by the data in Fig 6, at least in terms of making strong conclusions. My recommendation is that the authors temper their conclusions of the FRAP data by saying that "the tectonic complex *may* act as a diffusional gate regulator to sort membrane proteins", and that "depletion of Tctn1 *may* cause increased diffusion rates at the transition zone".

Reviewer #2 (Remarks to the Author):

The revised manuscript by Truong et al is markedly improved and the authors have addressed almost all of my concerns. There is only one point left:

My first point was:

In their work, Truong et al. aimed to "investigate how tectonic proteins regulate ciliary membrane protein composition" (lines 72 and 73). To reach this aim, they "generated a photoreceptor-specific Tctn1 knockout mouse" (lines 73 and 74). Undoubtedly, it is important to uncover the mechanism of how the Tctn proteins control ciliary gating and it is also crucial to investigate the function of proteins in the connecting cilium of photoreceptors. Considering that there is a structural and functional diversity of vertebrate primary cilia and that vertebrates present very specialised cell and tissue-specific types of primary cilia, I feel that the author's statements:

"We further show that membrane proteins moved faster through the transition zone illustrating that the tectonic complex acts as a physical barrier to slow diffusion of membrane proteins so they can be properly sorted by ciliary transport carriers." (lines 21-24), "Together, these results indicate that the structural elements of the transition zone remain largely intact when the tectonic complex is depleted." (lines 182 and 183), "Therefore, we conclude that the tectonic complex helps to impede passive diffusion of transmembrane proteins, so that they can be efficiently removed by the BBSome." (lines 249 and 250) and "With our findings, we can now appreciate that these ciliary alterations are due to a leaky diffusional barrier." (lines 263 and 264) are overinterpreted since their findings do not allow to draw a general conclusion but only a photoreceptor-specific. As mentioned by Truong and colleagues, "retinal photoreceptors contain a modified primary cilium" (line 69). This implicates that this type of cilium is different from other primary cilia. For this reason, it has to be shown that the findings in photoreceptor cilia are transferable to other types of primary cilia in order to formulate a general statement.

The authors answered as follows:

The role for tectonic and MKS proteins has been well studied in other types of primary cilium (PMIDs: 21763481, 21725307, 26540106, 21422230, 25869670, 22179047, 26392567, 26595381). By building off a rich history of research, we do not feel that we are overstating our findings as these proteins were first theorized to play a role in forming a membrane gate through loss-of-function studies that showed alterations in the ciliary membrane composition. FRAP experiments also found that loss of either B9d1 (a MKS complex protein) or septin2 (a cytoskeleton protein required for B9d1

localization, PMID 20558667) result in increased recovery of membrane proteins in the cilium. However, due to the limited resolution of the TZ in primary cilium these studies did not look directly at passive diffusion in the TZ membrane, so had to infer that recovery at ciliary membrane was faster due to alteration in the TZ. One unique feature of the photoreceptor cilium – extended TZ length – allows us to directly test passive diffusion within this membrane. As to Reviewer 2's comment that "their findings do not allow to draw a general conclusion but only a photoreceptor-specific", we feel that this is an unfair statement to reduce the impact of our work as "specialized" when the cilia field is dominated by model organisms such C elegans, Chlamy, and cultured cells all of which contain cilia with specialized functions. Moreover, human genetics and proteomic data suggest that the composition of the BB, TZ, INV, and axoneme are highly conserved between the primary and photoreceptor cilium.

In times of great efforts to differentiate between cilia of various cell types in order to explain cell type-specific differences in cilia biology and to get novel insights into the mechanisms underlying these differences, I do not feel that my statement reduces the impact of the authors' work. Quite the opposite, I think that it is important to be precise and to discuss the results in a cell type-specific context. The dissection of cell type-specific differences is an important endeavour in cilia biology to understand the variability of ciliopathy symptoms and to pave the way for the development of therapies against ciliopathies.

Reviewer #3 (Remarks to the Author):

The authors have provided explanations to all my previous questions and I have no further points that need to be addressed.

I only have two small comments:

-What could be the molecular explanation for the proximally restricted CEP290 staining in Tctn1 mutant cells?

-The additional data that were provided on Figure S8 are only partially convincing, as it seems that bleaching conditions only lead to modest extinction of the signal and signal reduction is not limited to the ROI? Nevertheless we can rely on the quantifications of Figure 6.

Reviewer #4 (Remarks to the Author):

In this revision, the authors responded to every of my concerns. This reviewer is satisfied with the majority of the responses but not to the use of Rho-GFP as a marker for passive membrane diffusion at CC in FRAP experiment. As previously pointed out, it's still controversial if ciliary entry of rhodopsin involves IFT. Specifically, using Rho-GFP knockin mice, Williams Lab has showed by FRAP that the recovery of Rho-GFP at CC is reduced in rod-specific Kif3a-/- (PMC3428073); and Pazour Lab showed that acute deletion of Ift-140 caused Rho mislocalization to the inner segment plasma membrane (PMC4173073), indicating Rho's passing through CC involves IFT. Given recent progress on the role of IFT-A complex in ciliary entry of membrane proteins which involves Tubby family proteins (PMC5350516 and PMID: 36462505) and previous finding of mislocalization Rho to the extracellular vesicles and inner segment PM in Tubby and TULP1 mutant mice, it's hard (and inappropriate) to ignore the possibility of IFTA-mediated OS entry of rhodopsin in rods. The authors stated that they tested GFP-TM but the expression level at CC is too low to do FRAP. In this reviewer's opinion, this technical difficulty doesn't justify the choice of Rho-GFP as a marker for passive membrane diffusion at CC. Furthermore, in their rebuttal, the authors proposed faster disc morphogenesis and RPE phagocytosis as the reason for the lack of increase of Rho and other OS membrane proteins in the proteomic data. Can the authors use the classical retina radioautography (used by Richard Young and others in 1960s, 1970s) to prove the suspected elevated OS turnover? Also, if there is increased disc

morphogenesis in KO, it may be worth checking if the OS length is increased during the OS formation stage, such as at P14.

REVIEWERS' COMMENTS

Reviewer #1 (Remarks to the Author):

I appreciate the efforts the authors have made to improve the FRAP data, which is very important for the manuscript's conclusion that the tectonic complex establishes a barrier that impedes diffusion of membrane proteins across the connecting cilium. For example, the authors tried experiments with a different OS membrane protein reporter (eg. GFP-TM); unfortunately, the signal intensity was too low at the photoreceptor CC to generate enough data. The data they did generate with the GFP-TM at the plasma membrane, whilst in line with the notion of the tectonic complex impeding diffusion at the CC, is circumstantial at best. The representative CC images of pre- and post-bleach Rho-GFP in the new Supplemental Figure 8 (note: this figure is incorrectly referred to as Fig S7 in the manuscript) is nice to see; however, it is very hard for me to be convinced that the level of signal recovery is sufficient to make strong conclusions. Also, I wonder why the control trace does not drop to the same extent as the iCre/Tect-flox trace immediately post bleach? One of the reviewers suggested assessing the diffusion barrier model in another system. I don't think this was unreasonable given the marginal nature of the data in Figure 6 (in my opinion), and the weight put on that data to derive a major conclusion. Thus, in the final reckoning, I remain unconvinced by the data in Fig 6, at least in terms of making strong conclusions. My recommendation is that the authors temper their conclusions of the FRAP data by saying that "the tectonic complex *may* act as a diffusional gate regulator to sort membrane proteins", and that "depletion of Tctn1 *may* cause increased diffusion rates at the transition zone".

Response:

Thank you, we have corrected the error in the Supplemental Figure numbering. Yes, we found that photobleaching GFP in thick fresh retinal sections was challenging and yielded variable depletion of the fluorescence signal. We appreciate the reviewer's concerns about drawing such strong conclusions from limited experimental data; therefore, have altered the manuscript to reflect a more "suggestive" model. As to their recommendation to assess whether the TZ diffusional barrier is disrupted by the loss of Tctn1 in another system is a good experiment to attempt in the future. However, we hope the reviewer considers that implementing cilia FRAP experiments in cell culture or another model organism within our laboratory - focused on mouse photoreceptor cell biology - is not trivial.

Reviewer #2 (Remarks to the Author):

The revised manuscript by Truong et al is markedly improved and the authors have addressed almost all of my concerns. There is only one point left:

My first point was:

In their work, Truong et al. aimed to "investigate how tectonic proteins regulate ciliary membrane protein composition" (lines 72 and 73). To reach this aim, they "generated a photoreceptor-specific Tctn1 knockout mouse" (lines 73 and 74). Undoubtedly, it is important to uncover the mechanism of how the Tctn proteins control ciliary gating and it is also crucial to investigate the function of proteins in the connecting cilium of photoreceptors. Considering that there is a structural and functional diversity of vertebrate primary cilia and that vertebrates present very specialised cell and tissue-specific types of primary cilia, I feel that the author's statements:

"We further show that membrane proteins moved faster through the transition zone illustrating that the tectonic complex acts as a physical barrier to slow diffusion of membrane proteins so they can be properly sorted by ciliary transport carriers." (lines 21-24), "Together, these results indicate that the

structural elements of the transition zone remain largely intact when the tectonic complex is depleted.” (lines 182 and 183), “Therefore, we conclude that the tectonic complex helps to impede passive diffusion of transmembrane proteins, so that they can be efficiently removed by the BBSome.” (lines 249 and 250) and “With our findings, we can now appreciate that these ciliary alterations are due to a leaky diffusional barrier.” (lines 263 and 264) are overinterpreted since their findings do not allow to draw a general conclusion but only a photoreceptor-specific. As mentioned by Truong and colleagues, “retinal photoreceptors contain a modified primary cilium” (line 69). This implicates that this type of cilium is different from other primary cilia. For this reason, it has to be shown that the findings in photoreceptor cilia are transferable to other types of primary cilia in order to formulate a general statement.

The authors answered as follows:

The role for tectonic and MKS proteins has been well studied in other types of primary cilium (PMIDs: 21763481, 21725307, 26540106, 21422230, 25869670, 22179047, 26392567, 26595381). By building off a rich history of research, we do not feel that we are overstating our findings as these proteins were first theorized to play a role in forming a membrane gate through loss-of-function studies that showed alterations in the ciliary membrane composition. FRAP experiments also found that loss of either B9d1 (a MKS complex protein) or septin2 (a cytoskeleton protein required for B9d1 localization, PMID 20558667) result in increased recovery of membrane proteins in the cilium. However, due to the limited resolution of the TZ in primary cilium these studies did not look directly at passive diffusion in the TZ membrane, so had to infer that recovery at ciliary membrane was faster due to alteration in the TZ. One unique feature of the photoreceptor cilium – extended TZ length – allows us to directly test passive diffusion within this membrane.

As to Reviewer 2’s comment that “their findings do not allow to draw a general conclusion but only a photoreceptor-specific”, we feel that this is an unfair statement to reduce the impact of our work as “specialized” when the cilia field is dominated by model organisms such C elegans, Chlamy, and cultured cells all of which contain cilia with specialized functions. Moreover, human genetics and proteomic data suggest that the composition of the BB, TZ, INV, and axoneme are highly conserved between the primary and photoreceptor cilium.

In times of great efforts to differentiate between cilia of various cell types in order to explain cell type-specific differences in cilia biology and to get novel insights into the mechanisms underlying these differences, I do not feel that my statement reduces the impact of the authors' work. Quite the opposite, I think that it is important to be precise and to discuss the results in a cell type-specific context. The dissection of cell type-specific differences is an important endeavour in cilia biology to understand the variability of ciliopathy symptoms and to pave the way for the development of therapies against ciliopathies.

Response: We agree with the reviewer that investigating cell type-specific differences in ciliary signaling and biology is a very impactful endeavor. We have removed the overinterpreted statement “With our findings, we can now appreciate that these ciliary alterations are due to a leaky diffusional barrier.” And have toned down the strong language we used when drawing conclusions about our work as requested by Reviewer 1 and the editor. We are happy that our own photoreceptor findings are in line with the previous data showing Tctn1 knockout in mice/MEFs/Celegans/Chlamy and feel this suggests a conserved function for Tctn1 in forming a TZ diffusional barrier to ensure proper protein composition of the ciliary membrane. Of course, which proteins are being sorted between ciliary and plasma membranes will absolutely depend on the cell type and so, how a cell ultimately responds to the loss of tctn1 will likely be cell or tissue-specific.

Reviewer #3 (Remarks to the Author):

The authors have provided explanations to all my previous questions and I have no further points that need to be addressed.

I only have two small comments:

-What could be the molecular explanation for the proximally restricted CEP290 staining in Tctn1 mutant cells?

-The additional data that were provided on Figure S8 are only partially convincing, as it seems that bleaching conditions only lead to modest extinction of the signal and signal reduction is not limited to the ROI? Nevertheless we can rely on the quantifications of Figure 6.

Response: Our MS data identified that CEP290 was downregulated in the Tctn1flox/iCre outer segments, and we believe this is reflected in the subtle basal shift in Cep290 localization. Considering that the TZ is composed of a dense network of ciliary proteins, it is difficult to pinpoint whether the driver of Cep290 mislocalization is directly due to the loss of Tctn1 or other affected proteins. We appreciate the reviewer's confidence in our FRAP results.

Reviewer #4 (Remarks to the Author):

In this revision, the authors responded to every of my concerns. This reviewer is satisfied with the majority of the responses but not to the use of Rho-GFP as a marker for passive membrane diffusion at CC in FRAP experiment. As previously pointed out, it's still controversial if ciliary entry of rhodopsin involves IFT. Specifically, using Rho-GFP knockin mice, Williams Lab has showed by FRAP that the recovery of Rho-GFP at CC is reduced in rod-specific Kif3a^{-/-} (PMC3428073); and Pazour Lab showed that acute deletion of Ift-140 caused Rho mislocalization to the inner segment plasma membrane (PMC4173073), indicating Rho's passing through CC involves IFT. Given recent progress on the role of IFT-A complex in ciliary entry of membrane proteins which involves Tubby family proteins (PMC5350516 and PMID: 36462505) and previous finding of mislocalization Rho to the extracellular vesicles and inner segment PM in Tubby and TULP1 mutant mice, it's hard (and inappropriate) to ignore the possibility of IFTA-mediated OS entry of rhodopsin in rods. The authors stated that they tested GFP-TM but the expression level at CC is too low to do FRAP. In this reviewer's opinion, this technical difficulty doesn't justify the choice of Rho-GFP as a marker for passive membrane diffusion at CC. Furthermore, in their rebuttal, the authors proposed faster disc morphogenesis and RPE phagocytosis as the reason for the lack of increase of Rho and other OS membrane proteins in the proteomic data. Can the authors use the classical retina radioautography (used by Richard Young and others in 1960s, 1970s) to prove the suspected elevated OS turnover? Also, if there is increased disc morphogenesis in KO, it may be worth checking if the OS length is increased during the OS formation stage, such as at P14.

Response: Two important points regarding previous literature.

First, in William's Kif3a^{-/-} paper they initially use hTERT-RPE1 cells to show Rho-GFP FRAP rates are slower in KIF3A knockdowns. Recently, William's lab published high-resolution SIM images showing that overexpression of Rho-GFP in hTERT-RPE1 cells does not localize to the ciliary membrane but is instead localized to the periciliary membrane of the ciliary pocket (PMC8425976). So, the original rates described in cell culture are actually assessing the delivery of Rho-GFP to the periciliary membrane.

When performing similar Rho-GFP FRAP experiments in mouse rods, William's lab did not utilize a marker to specifically identify the CC but attempted to identify this region through GFP fluorescence alone. These FRAP experiments produced rates that were ~3-fold slower than our own rates for Rho-GFP in WT mice. Our rates are in line with diffusion rate constants determined for the CC of Frog rods (PMC6829649). Given this discrepancy, we believe that the slow recovery rates reported in William's paper are not likely coming from Rho-GFP mobility within the CC but possibly from the delivery of Rho-GFP to the apical "periciliary" membrane.

Second, many studies have shown rhodopsin mislocalization from the outer segment in mutant mice and concluded that there is an impact on rhodopsin trafficking. However, it is not clear that any of these mutations represent a direct, causal relationship, especially considering that generally rhodopsin mislocalization occurs over time and is always incomplete in these mice. With regards to IFT-140 mutant mice, mistrafficking of rhodopsin could result from improper ciliogenesis, alterations in the homeostasis of the CC, or defects in delivery to the "periciliary" membrane (and not within the CC membrane). It is important to note that a recent BioRxiv paper suggests that normal trafficking of rhodopsin is via the inner segment plasma membrane (PMC10153271), so the presence of rhodopsin in the inner segment plasma membrane may not indicate mistrafficking. To date, no direct evidence has shown that rhodopsin is actively transported via IFT within the photoreceptor CC.

Finally, considering that the rate of passive diffusion is sufficient to deliver all the rhodopsin molecules into the outer segment and a more recent publication has found that rhodopsin is exclusively diffusing when heterologously expressed in the primary cilia membrane of IMCD3 cells (PMC5305262), we feel that Rho-GFP remains a reasonable marker for passive diffusion through the CC.

Assessing rates of outer segment renewal in the *Tctn1*^F/*iCre* mice is outside the scope of this current paper, although something that we do intend to attempt in the future. As to the reviewer's recommendation to compare outer segment length in P14 retinas. In the *iCre* mice, Cre expression turns on around P8. Without a *Tctn1* antibody, we cannot confirm that *Tctn1* is absent in rods at P14, and it would be expensive and time-consuming to perform MS experiments to do so. Since we cannot verify the loss of *Tctn1*, then we feel it is not worth assessing OS lengths at that age.